

# Hyper-resolution large-scale hydrological modelling benefits from improved process representation in mountain regions

Joren Janzing[1,2,3], Niko Wanders[4], Marit van Tiel[5,6], Barry van Jaarsveld[4], Dirk N. Karger[6], and Manuela I. Brunner[1,2,3]

[1]WSL Institute for Snow and Avalanche Research SLF, Davos Dorf, Switzerland
[2]Institute for Atmospheric and Climate Science, ETH Zurich, Zurich, Switzerland
[3]Climate Change, Extremes and Natural Hazards in Alpine Regions Research Center CERC, Davos Dorf, Switzerland
[4]Department of Physical Geography, Utrecht University, Utrecht, the Netherlands
[5]Laboratory of Hydraulics, Hydrology and Glaciology, ETH Zurich, Zurich, Switzerland
[6]Swiss Federal Institute for Forest, Snow and Landscape Research WSL, Birmensdorf, Switzerland

**Correspondence:** Joren Janzing (joren.janzing@slf.ch)

**Abstract.** Many of the world's major rivers originate in mountain regions and a large fraction of the global population relies on these regions for their water supply. The hydrological cycle of mountain regions and their dependent downstream regions are often studied using large-scale to global hydrological models (LHMs). The increasing spatial resolution of these models allows for improved representation of complex mountain topography, but existing model deficiencies in cold and high-elevation
regions limit potential model performance gains. Such model performance gains might be realized by investing into a better representation of hydrological processes that are relevant in mountain regions such as snow-accumulation and -melt. However, how much improved process representation would increase LHM performance remains largely unquantified. Here, we set up the hyper-resolution global hydrological model PCR-GLOBWB 2.0 (PCRaster Global Water Balance) over the larger Alpine domain and implement several changes to make it better suited at representing hydrological processes in mountain regions.
These changes include a.) the use of novel high-resolution meteorological forcing datasets; b.) an extended snow module based on a seasonally varying degree-day factor and an exponential melt function; c.) a regional calibration of the snow module against a snow reanalysis product; d.) a new integrated glacier module; and e.) increasing the contributions to the fast runoff components in the soil. Our evaluation of the effect of these different adjustments on model performance for discharge shows that while the meteorological forcing has a major effect on discharge simulations, its effect on performance is not
unidirectional over the domain. In addition, the structural and parametric changes, i.e. the snow module modification, glacier representation and runoff partitioning, improve discharge simulations in mountain regions: the snow module modification leads to an improved representation of the snowmelt peak for high-elevation catchments, the glacier module supplies additional water to glacierized catchments, and runoff partitioning in the soil improves the representation of streamflow in flashy catchments at lower elevations. We use these insights to present a new setup of the large-scale and hyper-resolution PCR-GLOBWB 2.0
model that is better suited to study hydrological processes in and beyond mountain regions around the world.



# 1 Introduction

Mountain regions play a critical role in supplying water to almost 2 billion people living in downstream regions and are therefore often referred to as the "water towers" of the world (Viviroli et al., 2007; Immerzeel et al., 2010, 2020). Water storage in snow and glaciers or a lack thereof is particularly important for drought development and recovery: the snow drought in the Italian Alps in 2022 developed into a streamflow drought downstream and affected many communities in the Po Plain (Colombo et al., 2023), whereas Alpine glaciers provided surrounding rivers with surplus melt-water during the 2003 Central European Drought thanks to a heatwave (Van Tiel et al., 2023). Hydrological processes in mountain regions thus have an over-proportional footprint well beyond mountain ranges. Therefore, considering hydrological processes that are particularly relevant in mountain regions such as snow accumulation or glacier melt is critical when studying hydro-systems in mountains and their dependent downstream regions.

Large-scale or global hydrological models (LHMs) are often used to study mountain hydro-systems and their dependent downstream areas (e.g. Viviroli et al., 2007, 2020; Khanal et al., 2021), but also to examine many other hydrological systems that exceed the scale of individual catchments. Example applications of such models include water resources (e.g. Dolan et al., 2021; Leijnse et al., 2024) and climate change impact assessments, such as those performed within the Inter-Sectoral Impact Model Intercomparison Project (ISIMIP; Warszawski et al., 2014). However, the coarse spatial resolution of many large-scale models – often tens of kilometers – limits the usefulness of their output for policymakers, who are often interested in regional- to local-scale information. This scale gap triggered a call for hyper-resolution, kilometer-scale models that are applicable at continental to global scales (Wood et al., 2011; Bierkens et al., 2015), which has been addressed by an increasing number of studies: proposed model solutions include the 1 km setup of the ParFlow model for the Contiguous United States (Yang et al., 2023) or the PCR-GLOBWB 2.0 model over Europe at a similar resolution (30 arcsec; Hoch et al., 2023), which Van Jaarsveld et al. (2024) recently used to perform a first global run at hyper-resolution.

At coarse spatial resolutions, there is ample evidence suggesting that LHMs do not accurately capture mountain processes: recent evaluations found that global hydrological and land-surface models show particularly poor performance at high elevations (Heinicke et al., 2024) and in cold climates (Gädeke et al., 2020; Hou et al., 2023) compared to other regions. One of the underlying issues is the extreme heterogeneity in mountain regions (e.g. topography, meteorology and soil types). Hyper-resolution models are expected to represent this heterogeneity better and could thus improve model performance in mountain catchments (e.g. for snow simulations; Malle et al. (2024)). However, to realize such increased model performance, the processes at play also need to be represented and parameterized with sufficient detail and accuracy. Part of the reduced performance in mountain regions can indeed be attributed to issues with process representation, as indicated by misrepresentations of both the volume and timing of snowmelt peaks as well as poor performance in basins where glaciers are not represented (Gädeke et al., 2020). Furthermore, neglecting glaciers or snow transport also leads to the formation of unrealistic "snow towers" at high elevations (Freudiger et al., 2017; Hoch et al., 2023). Several studies thus suggest that improving cryospheric process



representation should be a focus of further LHM development (Gädeke et al., 2020; Heinicke et al., 2024; Van Jaarsveld et al., 2024).

Many LHMs represent snow melt using a temperature-index model which relates melt to air temperature via a degree-day factor (DDF; Telteu et al., 2021). Even though full energy balance models are becoming more popular, especially at the catchment scale, temperature-index models remain widely used because they come at reduced computational demand, require minimal meteorological forcing and are accurate when calibrated (e.g. Hock, 2003; Magnusson et al., 2015). However, LHMs often use very simplistic temperature-index schemes e.g. by using only one constant DDF or by omitting calibration. Snow module comparisons at the catchment scale suggest that performance gains might be achieved by changes in the structure of the snow module (Girons Lopez et al., 2020). Proper evaluation of such improvements hinges on the availability of high quality reference data to compare how snow is represented by different snow module structures. Evaluations of snow processes in LHMs are often performed against global products representing snow water equivalent (SWE) or snow cover fraction derived from satellite measurements or reanalyses (e.g. Schellekens et al., 2017; Gädeke et al., 2020; Van Jaarsveld et al., 2024). Of the two, SWE is hydrologically the most relevant in regions with seasonal snow cover, but SWE reanalysis products often have too coarse spatial resolution (often 25 km or more) to be representative for mountain regions (Mortimer et al., 2020). In addition, the coarse model resolutions of LHMs themselves have prevented direct comparisons of SWE simulations with SWE estimates at individual snow measurement stations before the era of high-resolution modeling. Now, higher spatial resolutions of LHMs and new detailed regional SWE reanalysis products (e.g. Mott et al., 2023; Olefs et al., 2020) enable such direct SWE comparisons with the output of LHMs at a regional scale for mountain regions.

Whereas snow modules are present in most LHMs, many models have largely neglected glaciers (Gädeke et al., 2020; Telteu et al., 2021; Hanus et al., 2024). Glaciers can be an important additional water source during their melt season (e.g. the average glacier storage change contribution to total runoff near the river mouth for the rivers Rhone in August: 25%, Rhine in August: 7%, Danube in September: 4%, (Huss, 2011)) or during drought (Van Tiel et al., 2021, 2023). Including glaciers in hydrological models can thus potentially improve discharge simulations (Wiersma et al., 2022; Hanus et al., 2024), although such improvements will be limited to the summer months and to regions with a substantial glacier cover. Glaciers are represented in hydrological modelling in two ways, namely by a.) including an internal glacier module in the hydrological model ("integrated models") or b.) using the output from an external glacier model as the input to the hydrological model (i.e. one-way coupling the models; "coupled models"). An example of a coupled model is the setup created by Hanus et al. (2024), who used output from the Open Global Glacier Model (OGGM; Maussion et al., 2019) as the input to the Community Water Model V1.08 (CWatM; Burek et al., 2020). Similarly, Wiersma et al. (2022) coupled the Global Glacier Evolution Model (GloGEM; Huss and Hock, 2015) to PCR-GLOBWB 2.0. Alternatively, an example of an integrated model at the catchment scale is HBV-light (*Hydrologiska Byråns Vattenavdelning*) (Seibert and Vis, 2012; Seibert et al., 2018a), which calculates glacier mass balance, area evolution and runoff internally. The external glacier models used in coupled model set ups generally have more detailed process representation or more detailed calibration than would be feasible for integrated glaciers in LHMs. Still, we argue that integrated models can also have certain advantages over coupled models. First, integrated glacier modules are physically consistent with the surrounding model framework, which is not necessarily the case for externally coupled





glacier models. For example, large-scale glacier models may use different precipitation correction factors for each individual

glacier, which is inconsistent with the precipitation in non-glacierized gridcells. Furthermore, glacier geometries evolve over time and assumptions have to be made on how increases in the non-glacierized area are dealt with by the hydrological model (e.g. Hanus et al. (2024) assume the same relative glacier area change over all cells covered by a glacier, whereas in reality area changes mainly affect the glacier terminus). Second, integrated models can be more flexible: an integrated module can – in contrast to coupled models – be run simultaneously with the hydrological model and avoids the coupling steps related to

transferring data between the models. This could make integrated models easy to use when forcing them with an ensemble of meteorological forcing datasets.

Aside from snow and glaciers, rivers in mountainous or hilly regions often respond rapidly to local rainfall events leading to "flashy" discharge behaviour. These flashy responses are caused by the heavy precipitation, thin soils and steep slopes that characterize these regions and that makes these regions susceptible to floods (Weingartner et al., 2003). Simplifications in the

representation of soil processes and runoff production seem to limit LHM performance in flashier basins (Gharari et al., 2019). Generally, LHMs split the soil into a few layers that store and exchange water, but the exact details can vary significantly: each model has a different number of soil layers (e.g. CWatM: 3 layers (Burek et al., 2020); WaterGAP: 1 soil layer (Müller Schmied et al., 2021)) and these layers can have different thicknesses (e.g. CWatM: upper layer 5 cm thick (Burek et al., 2020), Water-GAP: soil is 0.1 up to 4 m (Telteu et al., 2021)). Interaction between these soil layers determines how water is partitioned over

different runoff processes. Most LHMs are not locally calibrated (Telteu et al., 2021), relying instead on a standard parameterization rather than a highly-calibrated local parameter setup that can potentially obscure structural deficiencies (Refsgaard and Storm, 1996; Andréassian et al., 2012). Without representing additional soil processes, we hypothesize that hyper-resolution LHMs can already realize further performance gains by reconsidering standard parameterizations (Hoch et al., 2023). For example, on steeper slopes the contribution of near-surface runoff components is larger (Weingartner et al., 2003). Changing

how water fluxes from the soil are partitioned across different processes that contribute to discharge (e.g. reduced groundwater recharge and increased saturation excess and interflow) could thus potentially capture more flashy behaviour. This could also improve the local relevance of these models, although their main focus will remain the larger catchments.

Generally, hydrological modeling is sensitive to the meteorological forcing dataset used as input (e.g. Raimonet et al., 2017; Tang et al., 2023; Gebrechorkos et al., 2024). For hyper-resolution LHMs, the horizontal resolution of the meteorological forc-

ing dataset is of particular importance as using too coarse meteorological forcing can severely reduce potential performance gains from moving towards hyper-resolution hydrological modelling (Hoch et al., 2023). High-resolution meteorological reanalysis products can be derived by downscaling coarser reanalyses products by exploiting statistical, physical or heuristic relationships or by using dynamically generated regional reanalysis products, both of which often outperform coarser global reanalyses products, e.g. in representing precipitation (e.g. Karger et al., 2021b; Keller and Wahl, 2021). Furthermore, high

resolution products represent temperature gradients with elevation in more detail, which can be important for snow modelling (Malle et al., 2024). In addition, regional reanalysis products also explicitly represent higher resolution atmospheric dynamics not present in the statistically or heuristically downscaled products. While Hoch et al. (2023) studied the effect of the spatial resolution of the meteorological forcing dataset (using statistical downscaling) on hyper-resolution LHM performance, it




remains to be assessed how the exact procedure of deriving data at higher spatial resolutions influences model performance.
Furthermore, despite improved resolutions, precipitation products in particular are known for large uncertainties over mountain regions (e.g. Isotta et al., 2015; Gampe and Ludwig, 2017; Bandhauer et al., 2022). It is thus important to assess how sensitive hydrological model performance in mountain regions is to the specific biases and large uncertainties of meteorological forcing datasets by using multiple input datasets.

While large-scale hydrological simulations at higher spatial resolution have become feasible thanks to increasingly available
computational resources, it is yet unclear by how much hydrological simulations can improve when combining such high-resolution models with improved process representation and the latest generation of meteorological forcing datasets. Therefore, we here aim to explore the effect of (1) improving snow and glacier representations, (2) changing runoff partitioning in the soil, and (3) using different meteorological datasets in PCR-GLOBWB 2.0 on discharge simulations over the larger Alpine region. We hypothesize that hyper-resolution LHM performance for discharge in mountain regions will increase by (H1) improving
the representation of mountain hydrological processes, such as snow and ice melt; (H2) reviewing standard parameterizations; and (H3) using dynamical dowscaled forcing products that include a representation of smaller-scale atmospheric dynamics compared to other forcing products. To test these hypotheses, we first assess how strongly discharge simulations are affected by the meteorological forcing chosen to drive the model. Second, we quantify the effect of structural changes in the model setup on model performance, namely by expanding the existing snow module and adding a new glacier module. Third, we
study the effect of parameter changes on model performance by calibrating SWE against a regional SWE reanalysis product and changing parameters controlling the volumes of soil compartments.

## 2 Methods

### 2.1 Model setup and study outline

We use the PCR-GLOBWB 2.0 model (Sutanudjaja et al., 2018) in the 30-arcsec setup developed by Hoch et al. (2023) (approx.
1 km at the equator, 650 m in longitudinal direction in the Alps). The model runs at a daily time step. PCR-GLOBWB 2.0 is a global hydrological model and contains different modules, which represent both natural processes related to vegetation, snow, soil, groundwater and river routing and anthropogenic processes such as human water use and irrigation. Here, we use a regional model setup (longitude: 3-18°; latitude: 43-51°) covering the Alps and the upstream parts of the catchments of four major Central European rivers (i.e. the Rhone, Rhine, Danube, and Po; Figure 1). We focus on the period 1990–2019, as all
forcing datasets are available for this time period and initial glacier volumes are often only available for around the year 2000.

We implement the different forcing datasets and model changes in a step-by-step manner and thus perform several model runs. A schematic overview and further details on the sequential model runs performed in this study are provided in Figure 2 and Table 1, respectively.



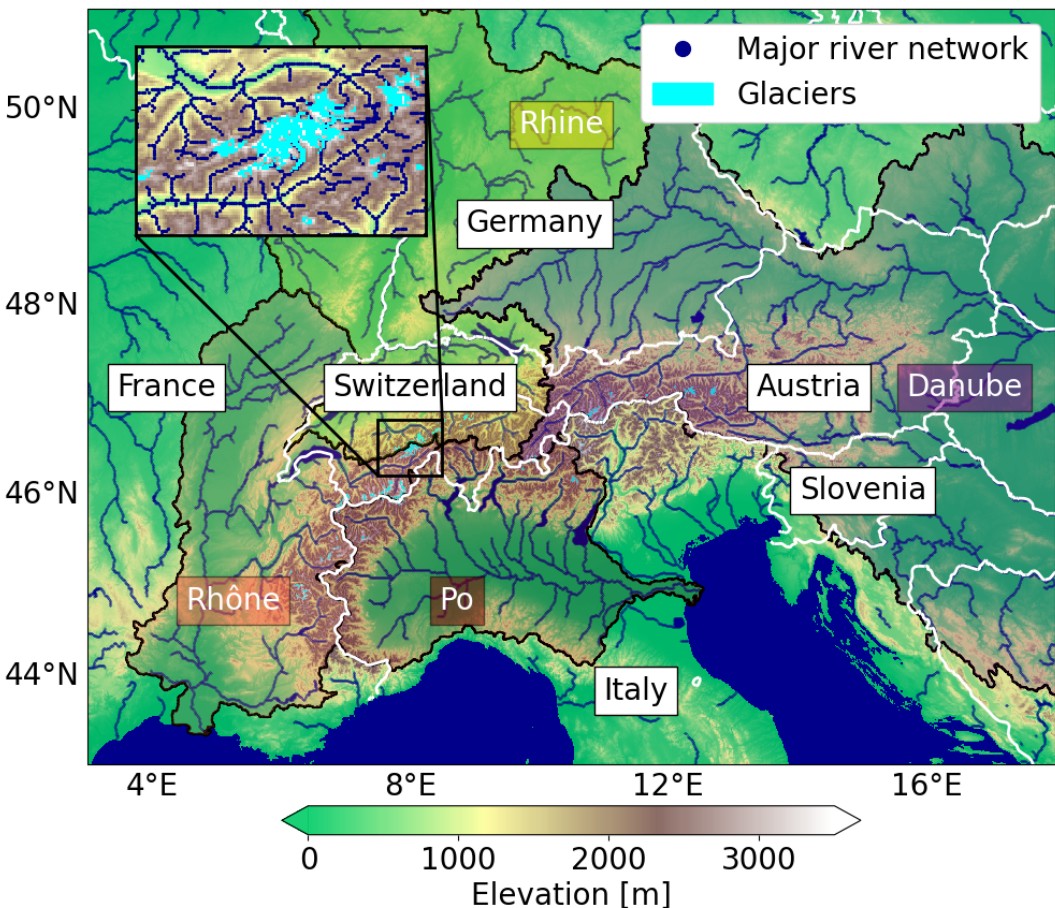

**Figure 1.** Overview of the model domain, highlighting the larger Alpine countries, major rivers and river basins, topography, and glaciers. The inset map shows the resolution of the model around the Aletsch Glacier (the largest glacier in the Alps). Elevations are derived from the upscaled MERIT Hydro DEM (Yamazaki et al., 2019).

## 2.2 Forcing and datasets

To quantify the sensitivity of hydrological models to the choice of the forcing dataset, we assess how discharge simulations vary under different meteorological forcing datasets. We focus on the input variables precipitation rate and near-surface air temperature as these are available for a wide range of potential meteorological datasets. Evaporation is then calculated within PCR-GLOBWB 2.0 using the method from Hamon (1963). For our comparison, we use the following meteorological datasets: (1) "STANDARD" input for hyper-resolution PCR-GLOBWB 2.0, (2) Climatologies at High resolution for the Earth's Land

Surface Areas v2.1 (CHELSA) and (3) Copernicus European Regional ReAnalysis (CERRA), further downscaled with the CHELSA algorithm (CERRA-CHELSA).



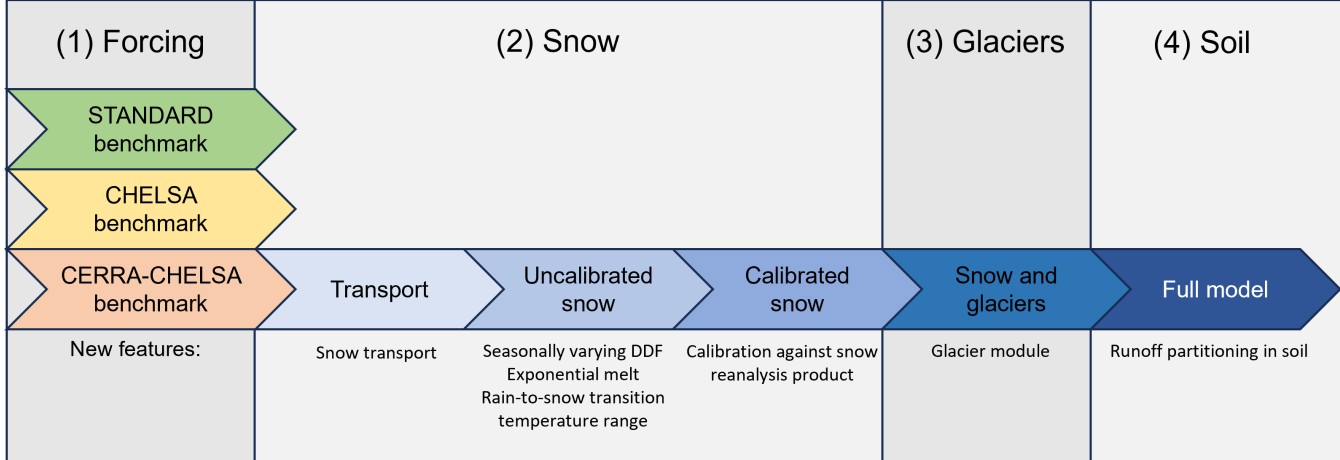

**Figure 2.** Overview of the different model runs performed in this study. (1) The standard model set-up with standard parameterization is our benchmark model and we run it with different forcing datasets. (2) Several changes are implemented in the snow module in different runs. (3) A new glacier module is added. (4) Runoff partitioning is adjusted, which is the final model change.

The STANDARD forcing was created by Van Jaarsveld et al. (2024). They created an internal meteorological downscaling scheme in PCR-GLOBWB 2.0. This scheme uses coarser scale meteorological input, in this case the W5E5 v2.0 (WFDE5 over land merged with ERA5 over the ocean) dataset (Lange et al., 2021), which has a spatial resolution of 0.5 degrees. This
coarser dataset is then downscaled to 30 arcsec spatial resolution using monthly climatologies from CHELSA-BIOCLIM+ (Climatologies at High resolution for the Earth's Land Surface Areas - bioclimatic variables plus) (Karger et al., 2017; Brun et al., 2022b). CHELSA (Karger et al., 2017, 2021a) is a downscaled reanalysis product based on the ERA5 product (Hersbach et al., 2020). CHELSA uses heuristic and physical relationships to downscale this forcing data to 30 arcsec spatial resolution. Downscaling is performed using topography, atmospheric lapse rates (for temperature; Karger et al. (2023)), spatial wind
fields, and the height of the boundary layer (for precipitation; Karger et al. (2021b)). Finally, CERRA (Schimanke et al., 2021; Ridal et al., 2024) is a regional reanalysis product over Europe provided at 5.5 km spatial resolution (approx. 180 arcsec). CERRA-Land is the associated surface analysis (Verrelle et al., 2022), which includes also additional data-assimilation of precipitation observations. We decided to use precipitation directly from CERRA-Land, since this dataset is already near the effective resolution of precipitation and the terrain effect could be over-represented by downscaling the data further (Daly
et al. (1997); Karger et al. (2021b) uses 3 km). In contrast, near-surface air temperature can be further downscaled to guarantee accurate spatial melt patterns at 30 arcsec resolution. Therefore, we created a new "CHELSA-CERRA" temperature dataset, for which temperature was taken from the CERRA dataset and was downscaled using the topographical CHELSA v2.1 algorithm (Karger et al., 2023). For simplicity, we refer to the combined meteorological product as "CERRA-CHELSA" in the remainder of this paper.
Within this study, we use several reference datasets against which we compare model inputs (forcing) and outputs such as streamflow, SWE, and glacier changes (Table 2). As a reference meteorological dataset, we use the Alpine Gridded Precipitation





| Run name | Snow | Glaciers | Soil | Forcing | Time period | Purpose |
|---|---|---|---|---|---|---|
| STANDARD benchmark | Standard, without snow transport | No | Standard | STANDARD | Evaluation II: 2010-2019 | Comparing meteorological input; Benchmark |
| CHELSA benchmark | Standard, without snow transport | No | Standard | CHELSA | Evaluation II: 2010-2019 | Comparing meteorological input |
| CERRA-CHELSA benchmark | Standard, without snow transport | No | Standard | CERRA-CHELSA | Evaluation II: 2010-2019 | Comparing meteorological input; Benchmark |
| Transport | Standard, with snow transport | No | Standard | CERRA-CHELSA | Evaluation II: 2010-2019 | Effect of snow transport |
| Uncalibrated snow | Updated, but uncalibrated | No | Standard | CERRA-CHELSA | Evaluation II: 2010-2019 | Effect of model structure changes |
| Calibrated snow | Updated | No | Standard | CERRA-CHELSA | Evaluation II: 2010-2019 | Effect of snow calibration |
| Snow and glaciers | Updated | Yes | Standard | CERRA-CHELSA | Evaluation II: 2010-2019 | Effect of glaciers |
| Full model | Updated | Yes | Updated | CERRA-CHELSA | Full period: 1990-2019 | Effect of runoff partitioning |

**Table 1.** List of the sequential model runs performed within this study.

Dataset (APGD; Isotta et al., 2014). This is a gridded product at 5 km spatial resolution covering the period 1971–2008 and is based on interpolated rain gauge data over the Alps. We choose this dataset as a reference, because unlike the other datasets it is specifically created for the Alps, does not use reanalyses, and has been used as a reference meteorological dataset before (Isotta et al., 2015). The dataset was however not corrected for undercatch (which can be tens of percentage points in the Alps (Sevruk, 1985)). Please note that we do not use this dataset as the forcing input, since LHMs are generally run at larger scales and therefore we used forcing products that are at least available at the continental scale.

Reference measurements of discharge in rivers around the Alps were taken from both national and regional agencies (sources listed in Table A1). We only selected stations in the basins of the Po, Rhine, Rhone and Danube rivers, leaving us with 2426 stations in total for the period 1990–2020. Please note that for evaluations, we only use stations that have at least 3 years of data over the considered evaluation period (see Section 2.4). Since this dataset is an extended and updated version of the





Large-sample hydro-meteorological dataset for the Alps (LHDA) from Schlemper et al. (2024), we refer to this dataset as the LHDA+ dataset. For each of these catchments, we derived a range of catchment characteristics. Catchment area and reservoirs were both derived from government agencies or derived from databases (see Table A1). Although our reservoir database is

more detailed than other large-scale databases such as the one by Lehner et al. (2005), it does not provide a complete overview of all reservoirs in the region. The fraction of the catchment covered by glaciers was computed from the Randolph Glacier Inventory 6.0 (Pfeffer et al., 2014; RGI Consortium, 2017). Snowfall fraction and potential evapotranspiration per catchment were calculated based on our own simulated model output.

For SWE comparisons, we use a set of different SWE products. We use two 1 km-gridded SWE regional reanalysis prod-

ucts at daily resolution, namely 1) a product by the Operational Snow Hydrological Service (OSHD) over (hydrological) Switzerland (Mott et al., 2023), which is available for the period 1998–2022, and 2) a product over Austria created with the SNOWGRID model (Olefs et al., 2013, 2020) for the period 1961–2020. Furthermore, we use the spatially much coarser SWE output from two atmospheric reanalysis products, namely CERRA-Land (5.5 km; 1984–2021) (Verrelle et al., 2022) and ERA5-Land (0.1 degree resolution (approx. 9 km); 1950–present) (Copernicus Climate Change Service, 2019). Finally, we

use data from 1047 Alpine measurement stations (251 in hydrological Switzerland) where SWE was inferred from snow depth (Fontrodona-Bach et al., 2023b).

For glaciers, we use spatially explicit maps on glacier elevation changes derived from satellite observations at 100 m spatial resolution (Hugonnet et al., 2021a), that are then upscaled to match the 30 arcsec model resolution. Here, we refer to this dataset as the "Glacier elevation change maps" (GECM). We use maps from two ten-year periods, namely 2000–2010 and 2010–2020.

We further use individual glacier mass balance measurements from the World Glacier Monitoring Service (WGMS) (World Glacier Monitoring Service (WGMS), 2023, version from 09-2023) and the Glacier Monitoring in Switzerland (GLAMOS) (GLAMOS - Glacier Monitoring Switzerland, 2022) datasets. We only considered glaciers that had at least six years of mass balance data over the study period and glaciers that have a minimum size of 3 km$^2$ (roughly 4-5 grid cells). Finally, we use glacier response time estimates from Zekollari et al. (2020b) (see Section 2.4) and glacier outlines and surface area estimates

from Raup et al. (2007) and GLIMS Consortium (2018).

For soil moisture, we use the European Space Agency Climate Change Initiative (ESACCI) COMBINED soil moisture data v8.1 (Gruber et al., 2019; Dorigo et al., 2017; Preimesberger et al., 2021). The ESACCI dataset includes satellite observations of soil moisture in the top 5 cm of the soil at a resolution of 0.25 degrees.

Finally, for analyses that use elevation, we use the Multi-Error-Removed Improved-Terrain Hydro Digital Elevation Model

(MERIT Hydro DEM) (Yamazaki et al., 2019). The MERIT Hydro DEM was upscaled from its original 3 arcsec resolution to 30 arcsec resolution by Hoch et al. (2023), who used it as the default DEM of the 30 arsec version of PCR-GLOBWB 2.0.

## 2.3 Model development

Based on the initial regional model setup introduced in Section 2.1, we further develop the representation of cryospheric and soil processes to improve discharge simulations in mountain regions. We aim to find a regionally valid setup that works well

for a larger domain and that is thus not directly fine-tuned for individual catchments, in line with the philosophy of many





**Table 2.** List of used datasets and their description.

| Name | Variables | Spatial coverage | Temporal coverage | Spatial resolution | Temporal resolution | Reference |
|---|---|---|---|---|---|---|
| **Meteorology** | | | | | | |
| STANDARD | Precipitation and temperature | Global | 1979-2019 | 30 arcsec | daily | Van Jaarsveld et al. (2024) |
| CHELSA (v2.1) | Precipitation and temperature | Global | 1979-2019 | 30 arcsec | daily | Karger et al. (2021a, b, 2023) |
| CERRA and CERRA-Land (for CERRA-CHELSA) | Precipitation and temperature | Europe | 1984-2021 | 5.5 km (temp. downscaled to 30 arcsec) | daily | Schimanke et al. (2021); Verrelle et al. (2022); Ridal et al. (2024) |
| APGD | Precipitation | Alps | 1971-2008 | 5 km | daily | Isotta et al. (2014); Isotta and Frei (2013) |
| **Discharge** | | | | | | |
| LHDA+ | Discharge, catchment attributes | Alps | 1990-2020 | Stations | daily | based on Schlemper et al. (2024) |
| **Snow** | | | | | | |
| OSHD | SWE | Switzerland | 1998-2022 | 1 km | daily | Mott et al. (2023); Mott (2023) |
| SNOWGRID | SWE | Austria | 1961-2020 | 1 km | daily | Olefs et al. (2020) |
| ERA5-Land | SWE | Global | 1950-present | 0.1 degree | daily | Copernicus Climate Change Service (2019) |
| CERRA-Land | SWE | Europe | 1984-2021 | 5.5 km | daily | Verrelle et al. (2022) |
| NH-SWE | SWE | Northern Hemisphere | 1950-2022 | Stations | daily | Fontrodona-Bach et al. (2023b) |
| **Glaciers** | | | | | | |
| GECM | Elevation changes | Global | 2000-2009; 2009-2019 | 100 m (upscaled to 30arcsec) | 10 years | Hugonnet et al. (2021a) |
| WGMS | Mass balance | Global | Varying | Glaciers | Yearly | World Glacier Monitoring Service (WGMS) (2023) |
| Consensus Estimate | Glacier volumes | Global | Varying, approx. 2003 | Varies per glacier (max. 200 m) | - | Farinotti et al. (2019) |
| Response times | Glacier response times | Alps | 2018 | | | Zekollari et al. (2020b, a) |
| GLIMS | Glacier outlines and area estimates | Alps | Varying | | | Raup et al. (2007); GLIMS Consortium (2018) |
| **Soil** | | | | | | |
| ESACCI | Soil moisture | Global | 1978-2023 | 0.25 degrees | daily | Gruber et al. (2019); Dorigo et al. (2017); Preimesberger et al. (2021); Dorigo et al. (2023) |
| **Elevation** | | | | | | |
| MERIT Hydro DEM | Elevation | Global | - | 3 arcsec (upscaled to 30 arcsec) | - | Yamazaki et al. (2019) |





global hydrological models. The next few sections describe the structural changes made to the PCR-GLOBWB 2.0 model. The parameters used in the equations are listed in Table S1. The calibration strategy for specific parameters is then further outlined in Section 2.3.4.

### 2.3.1 Snow module

The existing version of PCR-GLOBWB 2.0 includes a snow module consisting of a temperature index approach with a constant DDF. A temperature-index model generally has the following form:

$$M = \text{DDF}(T - T_{\text{thresh}}), \tag{1}$$

where $M$ represents the melt rate (m day$^{-1}$), DDF the degree-day factor (m°C$^{-1}$day$^{-1}$), $T$ the daily average temperature (°C), and $T_{\text{thresh}}$ the temperature threshold above which melt occurs. We build on this existing setup and expand it with elements of

the snow model outlined in Magnusson et al. (2014), namely (1) a seasonally varying DDF, (2) exponential dependence on temperature, and (3) a rain-to-snowfall transition temperature range.

First, we replace the constant DDF with a seasonally varying one to capture the effects of changes in the solar declination throughout the year, following the approach outlined in Slater and Clark (2006):

$$\text{DDF} = \frac{\text{DDF}_{\text{max}} + \text{DDF}_{\text{min}}}{2} + \sin\left(\frac{k2\pi}{366}\right)\left(\frac{\text{DDF}_{\text{max}} - \text{DDF}_{\text{min}}}{2}\right) \tag{2}$$

For the Northern Hemisphere, DDF$_{\text{max}}$ is the degree-day factor on 21st of June (summer solstice; m°C$^{-1}$day$^{-1}$) and DDF$_{\text{min}}$ is the degree-day factor on 22nd of December (winter solstice; m°C$^{-1}$day$^{-1}$). $k$ represents the day of the year since 21st of March (equinox; -). Second, we implement an exponential relationship between temperature and melt following Magnusson et al. (2014). This formulation makes the melt more sensitive to temperature than under the assumption of a linear relationship and allows for limited melt below the threshold temperature, to account for days when the average temperature is below the

threshold temperature, but the maximum temperature surpasses it.

$$M = \text{DDF}\, m_{\text{m}} \left( \frac{T - T_{\text{thresh}}}{m_{\text{m}}} + ln\left(1 + \exp\left(-\frac{T - T_{\text{thresh}}}{m_{\text{m}}}\right)\right) \right), \tag{3}$$

where $m_{\text{m}}$ is a parameter controlling the transition between melt and no melt (°C).

Third, we adapt the snowfall and rainfall partitioning to account for snow and rainfall coincidence by creating a temperature transition zone where rainfall smoothly changes into snowfall (Magnusson et al., 2014).

$$P_{\text{snowfall}} = \frac{P}{1 + \exp\left(\frac{T - T_{\text{snowfall}}}{m_{\text{p}}}\right)}, \tag{4}$$

where $P_{\text{snowfall}}$ represents precipitation falling as snow (m/day), $P$ total precipitation (m day$^{-1}$), $T$ daily average temperature (°C), $T_{\text{snowfall}}$ temperature below which most precipitation falls as snow (°C), and the parameter $m_p$ determines the range where snow and rainfall co-occur (°C). Aside from additions to the snow module, we also ignore refreezing in the snowpack, since previous analyses showed it did not improve simulations (Magnusson et al., 2014; Girons Lopez et al., 2020).



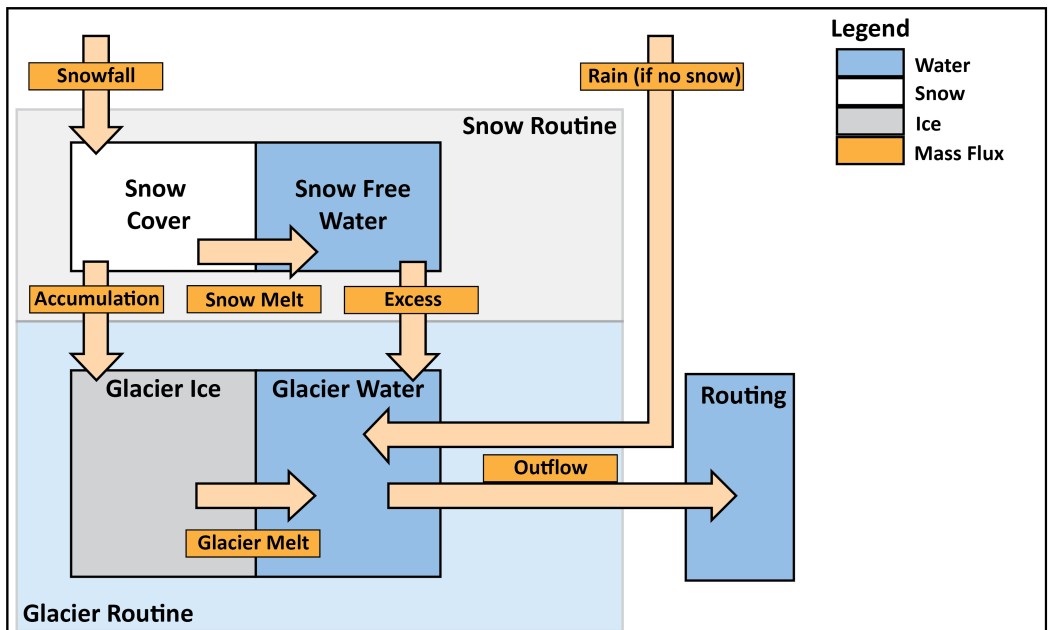

**Figure 3.** Schematic overview of the glacier module.

Furthermore, we also include the lateral snow transport scheme introduced by Van Jaarsveld et al. (2024) as a separate development step to better quantify its effect against the other development steps in the snow and glacier modules. Van Jaarsveld et al. (2024) implemented a lateral snow transport scheme based on Frey and Holzmann (2015) to avoid unrealistic snow accumulation at high elevations. This scheme transports part of the snow downhill based on the surface slope whenever the snowcover exceeds a certain SWE content. However, the transported snow is part of the glacier accumulation in many locations,

which is why we here apply the lateral transport scheme only outside of glaciers and define the accumulated snow on glaciers as glacier accumulation. When we introduce the glacier module, we thus restrict lateral snow transport to non-glacierized areas only. Note that this means that snow can be transported onto glaciers.

### 2.3.2 Glacier module

We introduce a new glacier module to PCR-GLOBWB 2.0. To create glacierized cells, we derive glacier geometries and

volumes from volume estimates by Farinotti et al. (2019), which are representative for the year 2003 for most glaciers. We then resample and regrid these volumes to the model raster of 30 arcsec, applying a correction factor to preserve the total ice volume of the glaciers. A cell is considered glacierized when any ice is present and there are no partially glacierized cells. The static part of the glacier scheme is based on Seibert et al. (2018a) and is schematically shown in Figure 3. The glacier consists of two parts: an ice reservoir and a water reservoir, representing water contained within the glacier. The glacier only melts when it is

not covered by snow, following a simple temperature-index scheme (see Equation 1) using the DDF for snow multiplied with a correction factor ($C_\mathrm{ice}$) to account for the higher albedo of the glacier ice surface (Seibert et al., 2018a). The glacier water



reservoir grows through the addition of glacier melt, snowmelt occurring on the glacier, and precipitation falling on the glacier during times when no snow is present. The water is then released from each individual glacier cell in the following way (Stahl et al., 2008):

$$Q = S(K_{\mathrm{min}} + K_{\mathrm{range}} * e^{-A_{\mathrm{g}}\mathrm{SWE}}), \tag{5}$$

where $Q$ is the glacial water release (m/day), $S$ the glacial water storage (m), and SWE the snow water equivalent on the glacier cell (m). $K_{\mathrm{min}}$, $K_{\mathrm{range}}$ (day$^{-1}$) and $A_{\mathrm{g}}$ (m$^{-1}$) are additional parameters determining the rate of melt water release.

For glacier accumulation, we deviate from Seibert et al. (2018a) and transfer all of the snow that is still covering the glacier on the 1st of September to glacier ice following Boscarello et al. (2014), instead of continuous conversion of a fraction of the snow to ice. The chosen approach facilitates comparisons with the OSHD dataset, which uses this date as the beginning of their hydrological year on which the model is reset to avoid unrealistic snow accumulation (Scherrer et al., 2024).

Glacier geometries change over time depending on their mass balance, which affects the quantity of melt over time. To account for such temporal changes, we implement the empirical $\Delta$h-parameterization scheme from Huss et al. (2010), as implemented in HBV (Seibert et al., 2018a). The $\Delta$h-parameterization (Huss et al., 2010) is based on the assumption that changes in glacier mass balance lead to specific patterns of change in glacier ice surface elevation. Huss et al. (2010) identified specific parameterizations for glaciers of different sizes that relate changes in mass to surface elevation changes. Note that we do not apply the width-scaling applied by Seibert et al. (2018a), as we are running the model in a spatially-distributed way. Before running the model, we create offline maps of distributed glacier thickness, where all glaciers lose a specific fraction of their mass (e.g. in steps of 1 percent mass loss) following Seibert et al. (2018a). During the hydrological model runs after the 1st of September of each year (i.e. after the glacier accumulation in our model), we update each individual glacier by reading in the distributed glacier thickness from these offline maps based on the mass balance simulated for the previous year. Mass changes do not necessarily occur in steps of 1 percent, leading to leftover mass or mass loss: for example, if total mass loss is 2.3 percent of the initial volume, we are left with 0.3 percent of leftover mass loss. To address this, we distribute such leftover mass or mass loss evenly over the glacier area. Under the $\Delta$h-parameterization, glaciers can only grow to their original extent. If glaciers gain mass compared to their initial extent, we add the additional mass to the gridcell downstream of the glacier. A full description of the $\Delta$h-parameterization is provided in the Appendix B.

### 2.3.3 Soil module

In the original model setup, each grid cell in PCR-GLOBWB 2.0 has two main soil layers and can contribute to river flow in three main ways: via direct runoff (infiltration or saturation excess), interflow, or groundwater contributions (Sutanudjaja et al., 2018). The first two components can be considered "fast" components, whereas the latter represents a "slow" component. Initial model runs performed with this standard setup suggested that the model produces too slow runoff responses and can not properly represent the fast components. Therefore, we introduced some measures to allow the model to produce more flashy runoff responses. Based on several model experiments, we decided to reduce the size of the top soil layer by making it half





as thick everywhere, while maintaining the total soil thickness constant. Since the maximum depth of the upper soil layer in PCR-GLOBWB 2.0 is by default set to 30 cm (Bierkens and Van Beek, 2009), halving the thickness still corresponds to a thickness of 15 cm, which is in line with the range of thicknesses that other LHMs are able to resolve (Telteu et al., 2021). This simple change should better represent the behaviour of thin soils often found on steep slopes (Weingartner et al., 2003), and makes saturation excess and interflow occur more rapidly and thus enables faster runoff responses.

### 2.3.4   Calibration

Although PCR-GLOBWB 2.0 is generally not calibrated, we do calibrate the degree-day factors of snow and ice to increase regional applicability and to ensure realistic glacier geometry evolution, since this evolution can be sensitive to biases in glacier mass balance. We calibrate on the specific process (e.g. on SWE) instead of discharge to avoid compensating with parameter calibration for deficiencies in other processes and to increase the stability of these parameters under varying temperatures

(Sleziak et al., 2020). We try to find one parameter set that is regionally valid. To save computational time, we perform the calibration of snow and glacier parameters offline, i.e. we run the snow and glacier component separately without running the rest of PCR-GLOBWB 2.0 and compare it to the reference datasets. We use dynamically dimensioned search (Tolson and Shoemaker, 2007) as implemented in the SPOTPY python package (Houska et al., 2015). In contrast, model evaluation is performed using the full model run. The considered calibration period is 2000–2009 (see Section 2.4).

The updated snow module required the calibration of 2 parameters, namely $DDF_{max}$ and $DDF_{min}$. As a reference dataset, we used a snow water equivalent reanalysis product over Switzerland (Mott et al., 2023). We chose this dataset because of its spatial continuity, its high quality which is unmet by products covering larger spatial domains (e.g. ERA5-Land or CERRA-Land), and because Switzerland covers diverse climatic regions. To explicitly account for elevation-dependent melt patterns, we averaged SWE spatially over partially-overlapping elevation zones (500-1500 m, 1000-2000 m, 1500-2500 m, 2000-3000

m, 2500-3500 m and Switzerland) and maximized the average Nash-Sutcliffe Efficiency (NSE) (Nash and Sutcliffe, 1970) across these elevation bands.

    After the snow calibration, we calibrate the glacier module. This module only required the calibration of one glacier correction factor for the DDF. We used satellite measurements of glacier elevation changes (Hugonnet et al., 2021a) as a reference dataset due to their coverage of all glaciers in the calibration domain (Switzerland for consistency with the snow calibration).

We calculated the elevation changes between the beginning and end of the calibration period (corrected for density differences between ice and water ($\rho_{ice} = 916.7$ kg m$^{-3}$ ; $\rho_{water} = 1000$ kg m$^{-3}$) and only focusing on locations with larger elevation changes (>2 m) to avoid noise) and minimized the mean absolute error for the elevation changes.

### 2.4   Evaluation

The aim of our evaluation is to assess the effect of the model development both on simulated discharge and the representation

of individual processes such as snow accumulation and melt. We thus evaluate the forcing datasets for precipitation and the model output for streamflow, SWE, glacier mass balance changes, glacier surface elevation evolution and soil moisture. We generally evaluate the daily simulations, except for the glaciers for which we use annual mass balances. For evaluation, we



split the study period into three blocks with different annual mean air temperature characteristics: (1) the calibration period (2000–2009), (2) a colder evaluation period (Evaluation I, 1990–1999); and (3) a warmer evaluation period (Evaluation II, 2010–2019) (see Figure 11A). The main evaluation is performed over Evaluation period II, which is also the period considered if a specific time interval is not specified. In Section 3.5, we use both evaluation periods to assess the transferability of the new model set up to different climatic regimes within the framework of a differential split sample test (DSST; Klemeš, 1986; Seibert, 2003).

The evaluation of the meteorological forcing focuses on precipitation, which is more spatially heterogeneous than temperature. We try to analyze the differences between the datasets and their realism, acknowledging the large uncertainties around high-altitude precipitation. We resample all three forcing datasets (STANDARD, CHELSA and CERRA; see Section 2.2) to the grid points of the coarser reference APGD and calculate both the Pearson correlation and the absolute bias against the APGD.

We evaluate the performance of the daily discharge simulations by comparing simulated discharge against observed streamflow, i.e. the station data from the LHDA+. We matched model grid cells to discharge stations by matching the catchment contours following Godet et al. (2024). We then evaluate discharge simulation performance by comparing the simulated to the observed time series using the Kling-Gupta efficiency (KGE; Gupta et al., 2009), using only stations that have at least three years of data over the considered evaluation period (number of valid stations: Evaluation period I: 1790; Calibration period: 2019, Evaluation period II: 2270). The KGE is defined as:

$$\text{KGE} = 1 - \sqrt{(r-1)^2 + (\frac{\mu_{sim}}{\mu_{obs}} - 1)^2 + (\frac{\sigma_{sim}}{\sigma_{obs}} - 1)^2}, \qquad (6)$$

where, $r$ is the Pearson correlation coefficient between observations and simulations, $\mu_{\text{obs}}$ is the mean over the observations, $\mu_{\text{sim}}$ is the mean over the simulations, $\sigma_{\text{obs}}$ is the standard deviation over the observations, and $\sigma_{\text{sim}}$ is the standard deviation over the simulations. KGE scores above -0.41 indicate that model simulations improve performance compared to assuming a constant flow corresponding to the average of the observed discharge time series (Knoben et al., 2019). To assess whether and by how much a specific change in model structure improves model performance for discharge, we use the KGE skill score (KGESS) as used by Knoben et al. (2020) and Van Jaarsveld et al. (2024), which compares the KGE score of a model run with a new setup against a model benchmark:

$$\text{KGESS} = \frac{\text{KGE}_{\text{model}} - \text{KGE}_{\text{bench}}}{1 - \text{KGE}_{\text{bench}}}, \qquad (7)$$

where, $\text{KGE}_{\text{model}}$ is the KGE score of the new model run and $\text{KGE}_{\text{bench}}$ is the KGE score of a benchmark model run, which represents an intermediate step within the model improvement chain. The Alpine region contains many reservoirs for water regulation, especially for hydropower production (Lehner et al., 2005; Brunner and Naveau, 2023), and their presence and how they are represented can influence model performance (e.g. Hanasaki et al., 2006; Abeshu et al., 2023). Therefore, we investigate how model performance for discharge varies among natural and regulated catchments. To separate regulated from natural catchments, we used a metric describing the deviation from a closed water balance (WB) assuming no long-term storage



effects (Salwey et al., 2023).

$$\mathrm{WB} = \frac{Q}{P} - (1 - \frac{E_\mathrm{P}}{P}), \tag{8}$$

where $Q$ is the (observed) averaged discharge (mm/day), $P$ is the precipitation (mm/day), and $E_\mathrm{P}$ is the potential evapotranspiration (mm/day) averaged over the catchment and study period. Salwey et al. (2023) have shown that this metric can be used to identify catchments affected by hydropower production and water transfers. As other factors such as errors in the meteo-

rological forcing or additional water input from glaciers can also lead to strong water balance deviations, we use this metric only as a rough indication for water transfers, hydropower production and other water balance deviations in combination with catchment-based information on reservoirs.

In addition to discharge, we evaluate model performance for SWE by calculating spatial averages over different elevation bands (0-1000 m, 1000-2000 m, 2000-3000 m or the entire country) both over Switzerland and over Austria. Then, we compute

the average seasonal SWE cycle over these zones and compare it to the seasonal SWE cycles derived from different snow reanalysis products (OSHD, SNOWGRID; see Table 2). Furthermore, we calculate the KGE for SWE by comparing modelled SWE and SWE inferred from observations at specific measurement stations (NH-SWE; see Table 2).

We evaluate glacier simulations both in terms of their mass balance and their geometry evolution. Simulated mass balances were summed over the hydrological year (starting in October to facilitate comparisons with the observations of WGMS and

GLAMOS; see Table 2) and compared to observed time series as well as to observed average mass loss. We only took time series with more than six data points over the Evaluation period II. To evaluate spatial patterns in glacier evolution, we visually compared spatial patterns of glacier surface elevation changes against observations (GECM; see Table 2). It is difficult to evaluate the long-term response of simulated glaciers to climate forcing given the relatively short study period. To address this problem, we repeat an experiment by Zekollari et al. (2020b), in which glaciers are continuously forced (for 300 years) with

the modelled mean mass balance for the period before 2018. The glaciers respond to this forcing by changing their shapes and they stabilize when they are in balance with the applied forcing. We then calculate the e-folding time scale (i.e. the time interval after which glaciers still had $1/e$ of their initial volume) and compare this to the estimates of Zekollari et al. (2020b).

We evaluate soil moisture against the ESACCI soil moisture dataset. This evaluation is more difficult than the evaluation of other variables, since the satellite data are not directly comparable to our model output: they measure soil moisture in the

top 5 cm of the soil, whereas we model moisture in the top 15 cm or 30 cm (see Table 2). Furthermore, the resolution of the observations is much coarser than the one of the simulations. Therefore, we resample our simulations to the resolution of the satellite data using spatial averages and only select locations with at least 50 percent of valid daily data over Evaluation period II. We calculate the Spearman rank correlation coefficient between the daily model simulations and the satellite observations to assess model performance, because this metric should be applicable despite the different soil moisture depths.





# 3 Results

## 3.1 Effect of meteorological forcing

Precipitation in all three meteorological forcing datasets used in this comparison correlate well with gridded precipitation station data from the APGD (see Figure 4). Precipitation from CERRA-CHELSA shows generally higher correlations with observed precipitation than the precipitation of the STANDARD and CHELSA input datasets (compare Figures 4A and B with C). The correlation between the three datasets and the reference dataset varies across the Alps and is especially low in the Po Plain. All meteorological datasets show a positive bias in precipitation over the Alps and a negative bias over the Apennines (see Figures 4D,E,F). Around the Alps, the STANDARD and CHELSA forcings show slightly positive biases, whereas CERRA-CHELSA shows a slightly negative bias. Overall, CERRA-CHELSA and CHELSA have a smaller precipitation bias than the STANDARD dataset (mean absolute bias: CERRA-CHELSA: 0.4 mm/day; CHELSA 0.5 mm/day; STANDARD: 3 mm/day).

The choice of meteorological forcing has a strong effect on simulated discharge, but the effect is not uniform across catchments. Using the STANDARD forcing, we see generally better model performance for discharge over the Alps than the surrounding areas (see Figure 4G). However, although performance is decent overall, locally there is poor model performance for discharge in certain catchments in the western Alps, southern Switzerland and eastern Austria. Forcing the model with CERRA-CHELSA leads to improved performance in these regions, as well as in southern Germany (see Figure 4H). In contrast, CERRA-CHELSA leads to reductions in model performance for discharge in eastern France, parts of Switzerland and western Austria. Using CHELSA leads to a slight worsening of model performance for discharge compared to runs with STANDARD or CERRA-CHELSA forcing, except in parts of the Alps (see Figure 4I).

In summary, we find that discharge simulations generated with the CERRA-CHELSA and STANDARD forcing datasets are generally better than those generated with the CHELSA dataset (see Figure 4J). As the precipitation of the CERRA-CHELSA dataset aligns better with the reference precipitation dataset than the STANDARD dataset (see Figure 4A, B, D and E), we performed all further analyses with the CERRA-CHELSA dataset.

## 3.2 Snow representation

SWE representation benefits to some degree from the proposed adjustments of the snow module (see Figure 5). The introduction of the snow transport scheme only improved snow representation at the highest elevations (2000–3000 m), where snow towers were a major issue (compare the Transport run with the CERRA-CHELSA benchmark run in Figure 5 C and F). Without calibration, further structural changes to the snow module (i.e. the seasonally varying DDF, exponential temperature dependence, and a rain-to-snowfall transition temperature range) lead to an improvement of the SWE representation at most elevations, except at the lowest elevations, where they lead to slightly too low melt rates (compare Uncalibrated snow and Transport runs in Figure 5A, B, D, and E). These differences between elevation zones suggest that the model structure is able to capture the elevation dependence of melt rates. Calibrating the DDF in the snow module against SWE leads to realistic but slightly too high melt rates over Switzerland (see Calibrated snow run in Figure 5A, B, and C). In Austria, the Calibrated snow run is, while slightly less accurate than in Switzerland, the most accurate of the presented model runs, even though no Austrian



**Figure 4.** Differences in meteorological forcing datasets and the effect of their choice on model performance for discharge over the larger Alpine domain. The top row compares the correlation of precipitation in the (A) STANDARD, (B) CHELSA, and (C) CERRA-CHELSA datasets with observed precipitation (APGD, 2000–2008). The middle row shows the bias of mean daily precipitation against observed precipitation for the (D) STANDARD, (E) CHELSA, and (F) CERRA-CHELSA datasets. The bottom row compares discharge simulations generated with the different forcing datasets: (G) KGEs for the STANDARD benchmark run. KGESS for the CHELSA benchmark run (H) and the CERRA-CHELSA benchmark run (I) against the STANDARD benchmark run. (J) Ridge line plots showing the distribution of KGEs (in number of catchments) for the three benchmark runs. Note that roughly 6 percent of stations have a KGE smaller than -1 and fall outside of the bounds in J.





SWE data was used for calibration (see Figure 5D, E and F). This demonstrates that the snow module is generally transferable
to other regions. The comparison of our model runs against SWE estimates at measurement stations (see Figures 5G and H)

confirms that SWE simulations profit from the introduction of an improved snow transport scheme, seasonally varying degree
day factors, and their calibration.

Changes in SWE performance are reflected in the performance of discharge simulations (see Figure 6). Snow transport
improves discharge simulations in the highest parts of the Alps, but has hardly any effect outside of the mountains (see Figure
6B). Structural changes to the snow module lead to an improvement of discharge simulations in catchments at the highest

elevations, but a worsening in catchments at lower elevations (see Figure 6C). Finally, calibrating SWE improves discharge
representation in most of the catchments that worsened from the structural changes, with a few exceptions in the Alps (see
Figure 6D). Figure 7A, B, and C illustrate that calibration mainly decreases model performance for discharge in catchments
with reservoirs and/or a negative water gap.

### 3.3 Glacier representation

The new glacier module captures the general behaviour of glaciers: both spatial patterns of glacier elevation changes and
temporal patterns of mass balances are roughly reproduced (see Figure 8A,B,C,D,E, and F), even though the model shows
biases for individual glaciers in mass balance (see Figure 8G) or retreat (see rapid retreat of Mer de Glace in Figure 8B).
Generally, mass balances are slightly better captured for larger (area > 8 km$^2$) than for smaller glaciers (see Figure 8G).
Figures 8H and I show the results from the equilibrium-experiment based on Zekollari et al. (2020b). We see that most larger

glaciers find an equilibrium over time, although there is significant commited mass loss with some glaciers collapsing over time
(Figure 8I). Overall, we end up with 40% committed mass loss in 2018, which was also found by Zekollari et al. (2020b). Our
modelling scheme thus captures the general behaviour of long-term glacier responses, with glacier retreat adjusting to a new
steady-state condition. However, on average glaciers respond relatively faster than in the reference dataset, both for the total
ice volume (Zekollari et al. (2020b): 33 years in 2018; here we have 24 years) and for the individual glaciers (Zekollari et al.

(2020b): on average 49 years; here 43 years). Again, larger glaciers show better performance than smaller glaciers (Figure 8H).
In conclusion, our model evaluation shows that the new glacier module works reasonably well for the total of alpine glaciers
(see Figure 8I) and for larger glaciers (see Figure 8G and H), while it can be significantly biased at the scale of individual or
small glaciers (see Figure 8G, H and I).

The addition of the glacier module mainly improves discharge simulations in highly glacierized catchments (see Figure 6E

and Figure 7F). The positive effect of glaciers is much less visible in catchments with a small snow fraction, although some
individual rainfall-dominated catchments also show a slight improvement in discharge simulations as a result of adding a glacier
module (see Figure 7E). Furthermore, our results indicate that in certain regions, especially around the Rhone river in south-
western Switzerland, discharge performance can decrease with the addition of glaciers (see Figure 6E). Such performance
decrease generally occurs in catchments with a negative water gap or reservoirs (see Figure 7F).



**Figure 5.** Snow representation for different elevation zones and regions. Average snow climatology over Switzerland (top row) and Austria (middle row) for grid cells at elevations between 0-1000 m (A and D), 1000-2000 m (B and E) and 2000-3000 m (C and F) derived from the gridded simulations and reanalysis products. The reference products are the OSHD reanalysis for Switzerland and the SNOWGRID product for Austria. The bottom row shows KGEs of SWE time series at different measurement stations of Fontrodona-Bach et al. (2023b) over Switzerland (G; 251 stations) and the full Alpine domain (H; 1047 stations) for different model runs and reanalysis products. For G and H, note that roughly 3 to 10 per cent of stations have a KGE < -1 (>30 per cent for CERRA-Land and ERA5-Land).



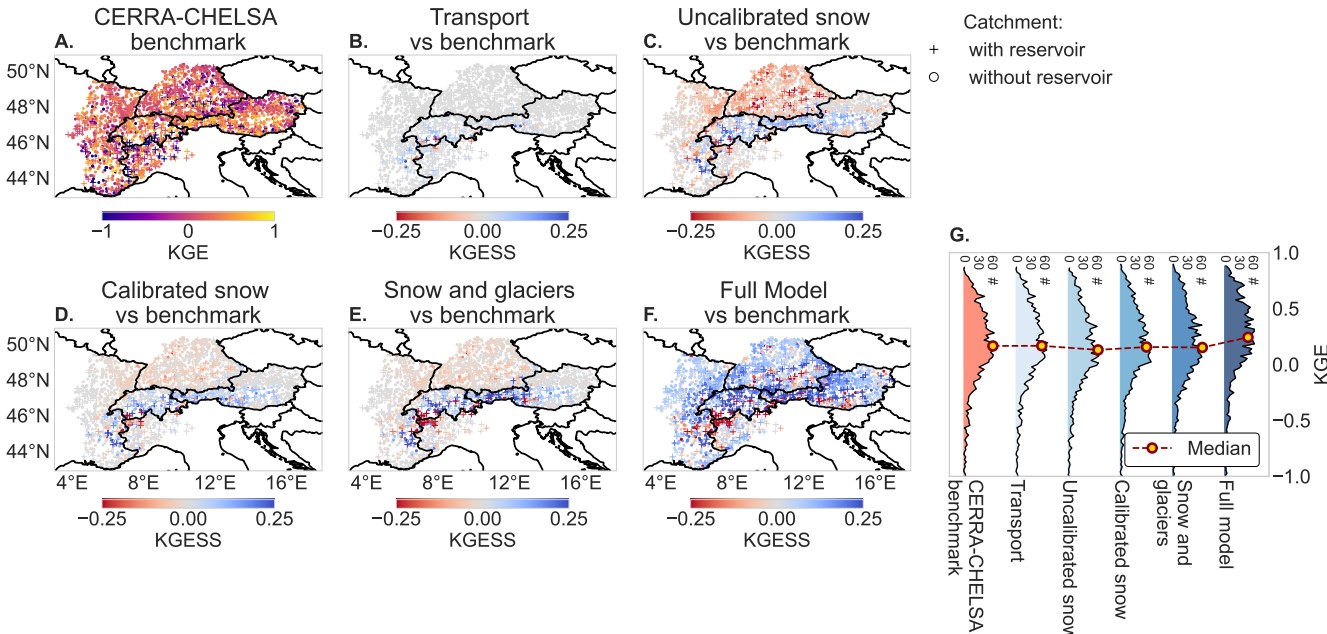

**Figure 6.** Model performance and its changes for discharge at the measurement stations for different adjustments in the snow routine. (A) Absolute KGEs for the CERRA-CHELSA benchmark run. Changes in performance (KGESS) with respect to the CERRA-CHELSA benchmark run for different model configurations: (B) Transport, (C) Uncalibrated snow, (D) Calibrated snow, (E) Snow and glacier and (F) Full model (including soil thickness change). (G) Distribution of KGE scores for the different model runs across catchments. Note that roughly 6 percent of stations have a KGE smaller than -1 and fall outside of the bounds.

## 3.4 Soil partitioning

Model performance for soil moisture varies over the domain (see Figure 9A and B), with generally higher performance in flatter low-elevation areas (such as the Rhone Valley, the Po Plain or the Rhine Valley) and lower performance over hilly or mountain areas. Implementing the soil changes has a mixed effect on performance: generally it improves performance in areas where the model was already performing well (i.e. the lower flatlands) and worsens performance in locations with lower model performance (see Figure 9C).

Over the entire domain, the changes made to the soil module, i.e. to the runoff partitioning, generally increase model performance for discharge (see Figures 6F). The improvements in discharge performance are strongest in catchments with lower snow fractions (<0.3; see Figure 7H) and are generally independent of the catchment area (see Figure 7G). Discharge performance slightly decreases in catchments with negative water balance gaps (see Figure 7G, H, and I).





**Figure 7.** Model performance change for discharge (KGESS) at the measurement stations after introducing different adjustments to the model. The model performance changes are compared to catchment characteristics. Top row: effect of the combined snow changes on discharge simulations (difference between the Calibrated snow and the CERRA-CHELSA benchmark runs); middle row: effect of the introduction of glaciers (Snow and glaciers vs. Calibrated snow); and bottom row: effect of the changes made to the soil (Full model vs. Snow and glaciers). (A), (D) and (G) show the dependence of model performance changes for discharge on the water gap and catchment area; (B), (E) and (H) on the water gap and snowfall fraction; and (C), (F), and (I) on the water gap and glacier area fraction. Note that roughly 3 percent of stations have a water gap larger than 1 and fall outside of the figure bounds.

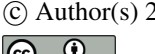



**Figure 8.** Evaluation of the glacier scheme newly integrated in the hydrological model. Top row shows the elevation change over 2010–2019 for the region around Glacier d'Argentière (A. observations; B. model) and around the Aletsch Glacier (C. observations; D. model). Observations from Hugonnet et al. (2021a), outlines from GLIMS Consortium (2018). (E) and (F): Annual time series of simulated against observed mass balance in meters water equivalent per year (observations Aletsch: GLAMOS - Glacier Monitoring Switzerland (2022); Argentière: World Glacier Monitoring Service (WGMS) (2023)). (G) Comparison of modelled mean mass balance (2010–2019) of small and large glaciers against observations in meter water equivalent per year (16 glaciers; World Glacier Monitoring Service (WGMS) (2023)). Finally, the response of glaciers to continuous forcing with the mean mass balance from 1990–2018 (see Section 2.4). (H) Comparison of e-folding response time to modelled estimates from Zekollari et al. (2020b) (136 glaciers). (I) Evolution of glacier volume over time under this continuous forcing.





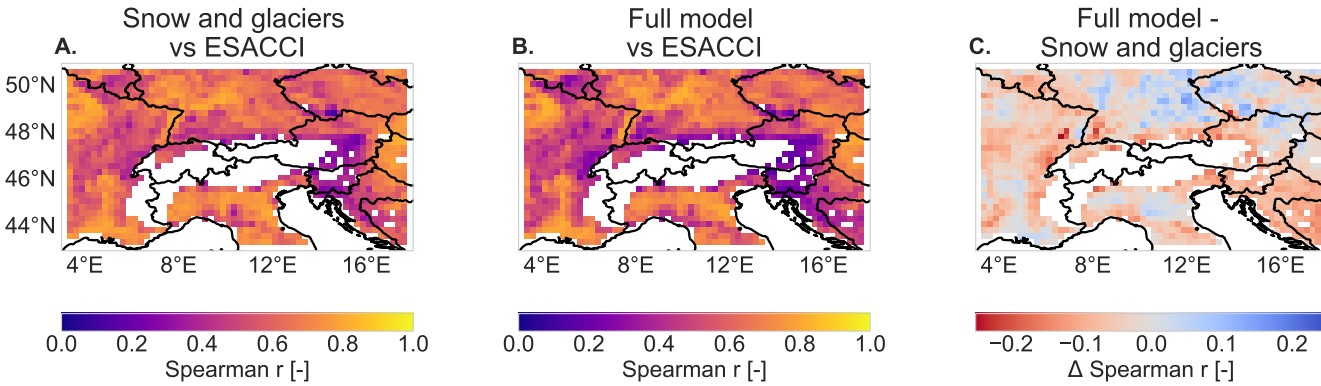

**Figure 9.** Comparison of soil moisture simulations against the ESACCI satellite observations. We show the Spearman correlation between the Snow and glacier (A) and Full model runs (B) against the ESACCI observations. (C) Difference in Spearman correlation coefficients between the Full model and the Snow and glaciers run. White areas indicate regions which had too many days without data (more than 50% of the time).

**Table 3.** Summary of the effect of the different model implementations on discharge simulations. The symbols have the following meaning: ↑: improvement, ↑↑: large improvement, ↓: worsening, ↓↓: large worsening, -: no effect, ∼: substantial but mixed effect.

| Catchment type | Meteorological forcing | Snow module | Glacier module | Soil partitioning |
|---|---|---|---|---|
| Small catchments | ∼ | - | - | ↑ |
| Large catchments | ∼ | - | - | ↑ |
| Natural rainfall dominated catchments | ∼ | - | - | ↑↑ |
| Natural snowfall dominated catchments | ∼ | ↑ | ↑ | ∼ |
| Natural glacierized catchments | ∼ | ↑ | ↑↑ | - |
| Regulated catchments | ∼ | ↓ | ↓ | - |

## 3.5 Evaluation of new model setup and its transferability in time

Our new model setup, which includes updated snow, glacier and soil modules, leads to general performance increases in streamflow simulations compared to the existing PCR-GLOBWB 2.0 setup, with performance depending on catchment characteristics (see Figure 10 and the summary provided in Table 3 and Figure 12). Catchment area and the water balance gap are major controls of absolute model performance for discharge in terms of KGE, which is highest in large and natural catchments, in which the water balance is nearly closed (see Figure 10A). Similarly, the model performs well in snow covered and glacierized catchments, where the model additions lead to a substantial improvement in model performance for discharge (see Figure 10B, C, E, and F). In contrast, the model adjustments can decrease performance in catchments with a water balance gap (see Figure 10D, E, and F).




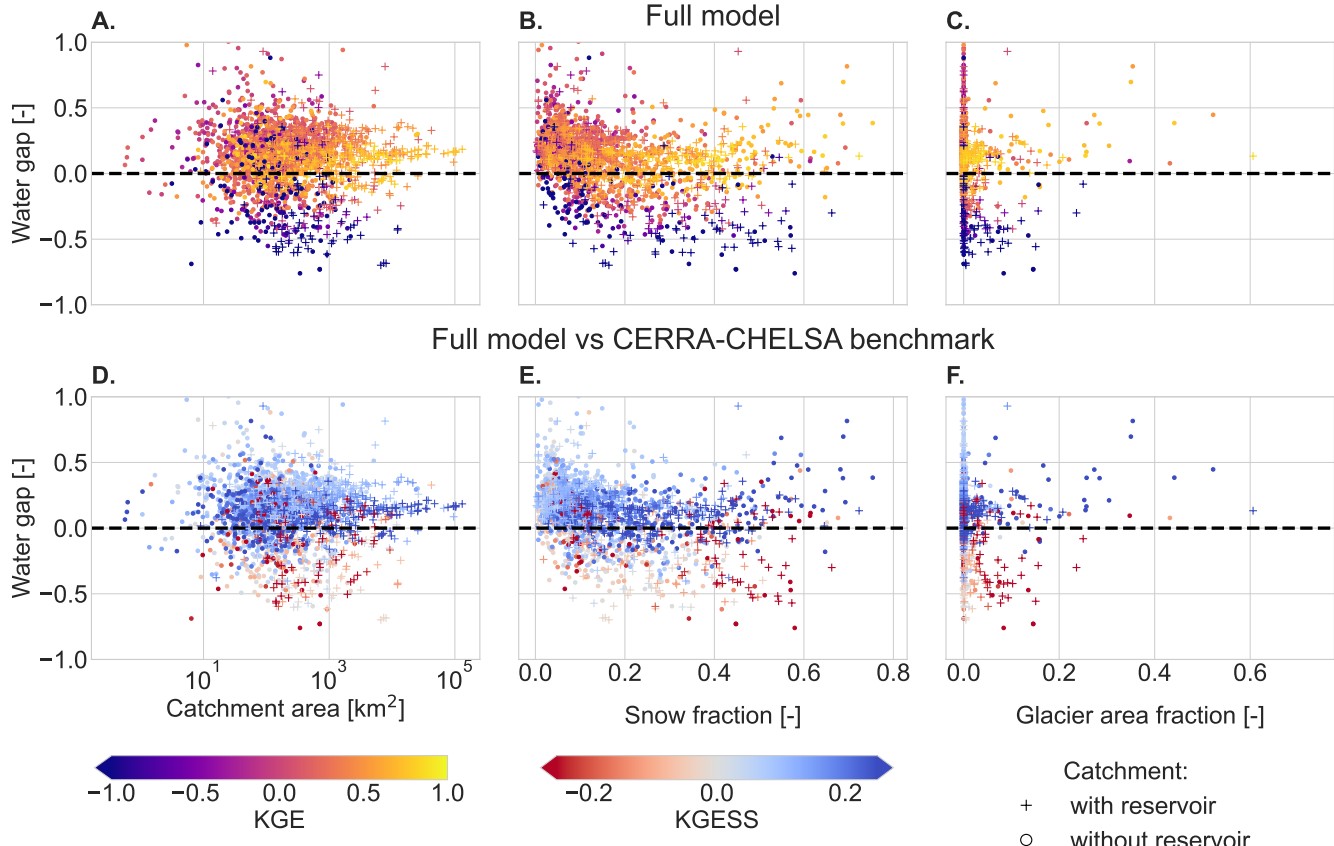

**Figure 10.** Model performance of the Full model run for discharge (top row) and the total model performance changes for discharge compared to the CERRA-CHELSA benchmark run (bottom row) in the evaluation catchments in relation to different catchment characteristics: water gap and catchment area (A,D), water gap and snowfall fraction (B,E), and water gap and glacier area fraction (C,F). Note that roughly 3 percent of stations have a water gap larger than 1 and fall outside of the figure bounds.

Finally, we study the extent to which model performance for SWE and streamflow simulations remains stable under varying
climate conditions, i.e. for periods with low vs. high average temperature. Model performance for discharge and SWE differs between the two evaluation periods representing different average temperature conditions, with performance change depending on the region (see Figure 11). The simulated average snow conditions over Switzerland still match observations very well over both evaluation periods (see Figure 11C). In contrast, simulated discharge performance differs for the calibration and evaluation periods (see Figure 11D and E). However, performance of discharge simulations can both increase and decrease
during the evaluation period as compared to the calibration period depending on the catchment. These changes have similar magnitudes as the changes due to different forcing datasets (compare to Figure 4G,H, and I). Overall, the warmer Evaluation II period shows a slight drop in median performance of streamflow simulations compared to Evaluation I period (see Figure





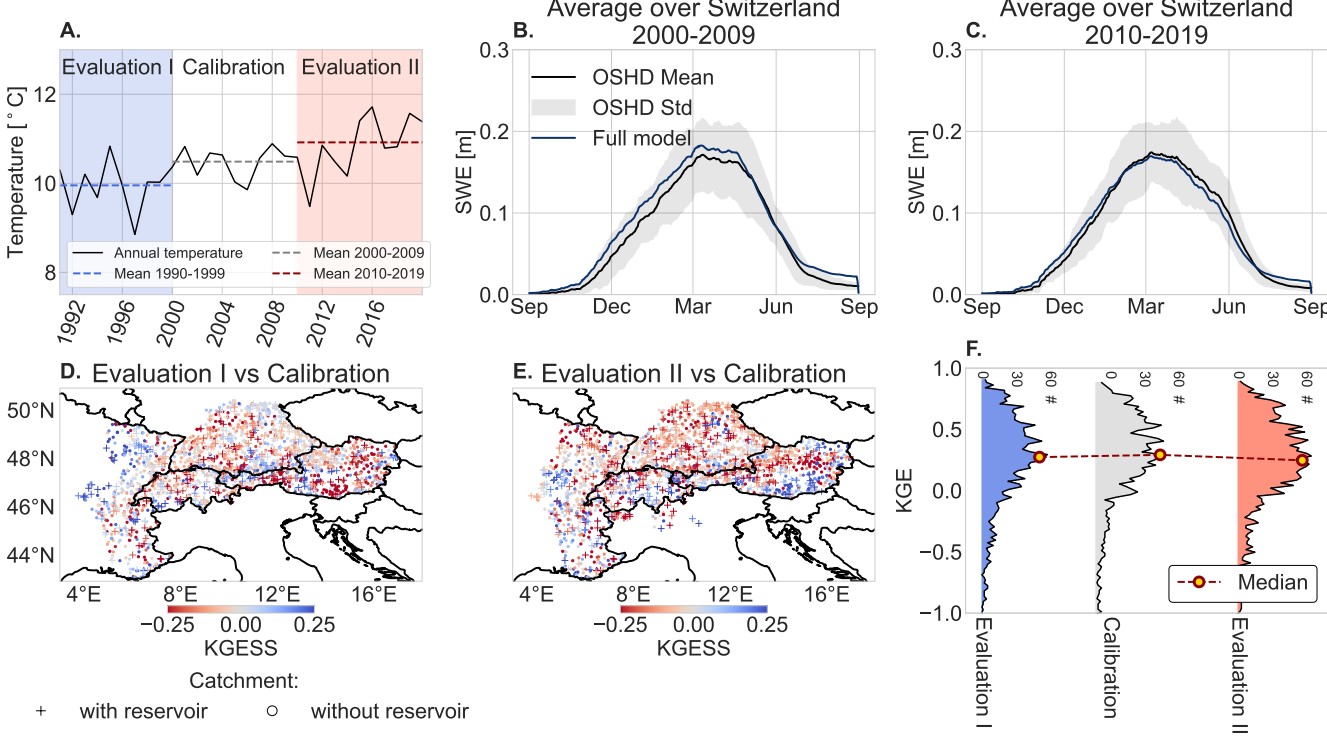

**Figure 11.** Dependence of model performance of the Full model run on time period and average temperature climatology. (A) Mean annual air temperature during the two evaluation and the calibration periods. Average climatology of snow over Switzerland for (B) 2000–2010 and (C) 2010–2019. Comparison of the performance of discharge simulations (KGESS) for (D) Evaluation period I and (E) Evaluation period II against the one in the calibration period. (F) Distribution of KGE scores over the three periods across the evaluation catchments.

11F). This change is, however, small, indicating that performance remains stable over time and is only weakly influenced by changing climate conditions such as increasing temperatures.

# 4 Discussion

## 4.1 Evaluation and recommendations for further model development

Our new model setup generally led to increased model performance compared to the old setup: structural and parameter changes applied to the snow, glacier and soil modules improved both SWE and discharge simulations (Figures 6, 7, 5, and 10). This highlights the importance of improving process representation in hyper-resolution modelling efforts.

Updating the runoff partitioning in the soil leads to a clear improvement in the simulation of discharge in natural catchments (Figures 6F and 7G,H, and I; column "Soil partitioning" in Table 3), with rainfall-dominated catchments (≈ less than 30% snowfall) benefiting the most (Figure 7G,H). This major increase in model performance is caused by a modest change in the





soil parameters, supporting the suggestion that the move to hyper-resolution requires careful review of parameterizations in LHMs (Hoch et al., 2023). Still, absolute performance in smaller rainfall-dominated catchments remains relatively limited

(Figure 10A and B). In addition, soil moisture representation did not improve in regions that already had poor performance (Figure 9C and D). Please note that the reference product used for computing model errors can have its own biases (Dorigo et al., 2015, 2017) and error estimates might therefore not be entirely representative. The representation of soil moisture and fast discharge responses thus needs to be further improved if LHMs are supposed to be applicable at smaller spatial scales. These further improvements can come from two directions: first, soil heterogeneity could be included in models in more detail.

For example, Van Jaarsveld et al. (2024) suggested that sub-grid scale land cover variability could still be important in hyper-resolution modelling and we hypothesize that this would also improve the representation of spatial variations in discharge behaviour. However, including this information would come at the cost of increased computational demands. Second, more explicit consideration of hill-slope processes such as preferential flow in the model structure (e.g. Rahman and Rosolem, 2017; Gharari et al., 2019; Fan et al., 2019) could lead to further improvements in the simulation of flashy runoff responses.

Discharge simulations also benefit from the structural changes made to the snow module, especially in snow-dominated catchments (Figures 6C and 7E; column "Snow module" in Table 3). Similarly, Girons Lopez et al. (2020) found for the catchment model HBV that exponential melt combined with seasonally varying DDFs improved discharge simulations, whereas the rain-to-snow transitions improved SWE representation but led to slightly poorer results for discharge. However, whereas our discharge simulations in most snow-dominated mountain catchments improved due to these structural changes, we noted

a slight decrease in discharge performance at lower elevations where snow contributions are less important. One reason for why the structural changes to the snow module are better suited for catchments in mountainous terrain might be differences in dominant snow processes between high and low elevations, such as the frequency of rain-on-snow events. Since Magnusson et al. (2014) built their snow model for alpine Switzerland, they might have prioritized representing melt patterns at higher elevations with thick snow cover over patterns in flatter terrain with only limited snowfall. For example, our scheme does not

explicitly include melt due to liquid precipitation. Another reason for this slight decrease in discharge performance at lower elevations could be related to our choice of regionally-averaged DDFs, since in reality these DDFs show smaller-scale variability in space (e.g. with aspect, albedo, elevation,..) (e.g. Hock, 2003; Ismail et al., 2023). Such variability is not accounted for in our model since we focused on regionally valid parameterizations instead of local solutions. However, ignoring this variability could lead to biases in SWE representation in specific locations. More elaborate snow module formulations, such

as parameterizations that include aspect (e.g. Immerzeel et al., 2012) or radiation (e.g. Hock, 1999), could increase our ability to capture more detailed spatial melt patterns. However, they come at the cost of increased model complexity and a larger number of input variables. In any case, the slight decrease in discharge performance at lower elevations was reduced via the calibration of modelled SWE against a regional SWE reanalysis, leaving us with a good overall discharge and SWE simulation performance (compare Figure 6C and D). The improved discharge representation after calibration highlights the performance

gains that can be achieved by including more regional data into LHMs. However, regional calibration is only possible due to the comparably high quality of observational and reanalysis data in Switzerland. Many other regions around the globe remain faced with a lack of observations of water balance components (Wilby, 2019), which can make accurate regional calibration



challenging. Furthermore, the slightly reduced performance in SWE representation over Austria (which was excluded from calibration) compared to Switzerland already indicates that parameters can vary between regions (compare Figure 5D,E,F with

Figure 5A,B,C). Highly-detailed calibration in one region might give a false sense of accuracy when applying the set-up outside of the calibration region. Regionally valid datasets must thus be chosen with care .

    Adding a glacier module led to general improvements in discharge simulations in glacierized catchments in the Alps, especially in catchments with a negligible or positive water gap (i.e. more observed discharge than is "expected"; see Figure 7F;

column "Glacier module" in Table 3). The effect of glaciers on general discharge performance further downstream remains limited (see Figure 6E), although this effect might be larger for certain months or seasons (Wiersma et al., 2022). Aside from discharge, glacier mass balances and spatial patterns in elevation changes are also reasonably well represented (see Figure 8). Still, we note that individual and especially smaller glaciers can show significant biases in the mass balance (see Figure 8G) and in the response times (see Figure 8H). The generally shorter response times than in previous experiments might be partly ex-

plained by the Δh-parameterization, which ignores potential delays in glacier response to mass changes (Seibert et al., 2018a). Further biases in both mass balance estimates and response times are likely related to the relatively coarse spatial resolution of our model compared to the size of individual glaciers, which makes it more difficult to accurately describe patterns in melt or snow accumulation for small glaciers consisting of only a few grid cells. Melt representation of individual glaciers could be improved by resolving glaciers at higher spatial resolution than the one of the LHM, for example by including elevation zones

(Seibert et al., 2018a). Resolving glaciers at higher spatial resolution could also lead to even more realistic glacier retreat, since the Δh-parameterization was originally designed for higher spatial resolutions (Huss et al., 2010) and is likely less accurate at the coarser 30-arcsec model grid. Further gains in glacier representation could be realized by improving glacier accumulation estimates by further developing the snow component, or by improving spatial melt patterns by varying the glacier DDF in space (analogously to what we suggested for snow).

While the improvements in process representation generally lead to an increase in model performance for discharge, there are regions where the performance of discharge simulations decreases after implementing the structural and parametric changes (see Figure 10D,E,and F). Our analysis shows that such performance decreases are common in catchments with gaps in the water balance, which can be related to issues with the meteorological forcing, glacier melt estimates or with the representation of water abstractions or (hydropower) reservoirs in the model. Indeed, the strongest negative performance changes after the

introduction of snow and glaciers often occur in catchments with reservoirs (see Figure 7B and F; row "Regulated catchments" in Table 3). In the Alps, significant glacier or snow melt occurs above hydropower reservoirs (e.g. in Switzerland, 4% of all hydropower is related to glacier mass loss (Schaefli et al., 2019)). Hydropower severely changes flow seasonality (Arheimer et al., 2017), essentially decoupling observed streamflow from snow and ice melt. Accurate representation of river regulation and hydropower reservoirs is thus important, but LHMs appear to have difficulty with modelling streamflow in these regulated

catchments (Veldkamp et al., 2018; Tu et al., 2024, e.g.). These issues might become even more apparent at hyper-resolution, because a.) small rivers are now represented and might be affected by reservoirs that are not represented in the model, and b.) reservoir schemes in LHMs were developed to represent regulation behaviour on a coarse grid (e.g. they mimic the combined





effect of all reservoirs within a 50 by 50 km grid cell) and likely need an update when moving to higher spatial resolutions
(Shin et al., 2019). However, even if more reservoirs were introduced to the model, accurate discharge modelling in regulated

systems likely remains challenging due to limited data on operation strategies and regulations (Turner and Voisin, 2022). Thus,
both improving the representing of human water management in models and collecting new data should remain an active area
of research.

Besides model structure and parameterization, the meteorological reanalysis forcing has a major influence on model perfor-
mance for discharge (see Figure 4G,H, and I; column "Meteorological forcing" in Table 3). This implies that improvements in

simulation performance that can potentially be achieved by structural changes are bounded by the quality of the meteorological
forcing dataset, especially in a non-calibrated model set-up. Replacing the STANDARD meteorological forcing product with
CERRA-CHELSA led to an overall increase in correlation with the reference dataset (compare Figure 4A and C), but showed
mixed performance changes for discharge over the model domain (compare Figure 4G and I). This apparent inconsistency
could be related to uncertainties affecting our precipitation evaluation and the LHM. Our precipitation evaluation is likely

affected by uncertainties in the reference APGD: although the APGD is based on a particularly dense network of direct obser-
vations, it still has uncertainties related to precipitation undercatch (for which it was not corrected), and in regions with less
dense station coverage (Isotta et al., 2015). Our evaluation of the precipitation products should therefore be interpreted as an
indication of consistency with interpolated surface measurements but not necessarily as an indication of absolute performance.
Collecting new, better, and more observations of meteorological variables over mountainous regions should thus remain a pri-

ority. The limited performance for discharge in small flashy catchments, as was pointed out before, might further prevent the
direct translation from more accurate precipitation input to improved discharge performance. The ever-present uncertainty in
forcing datasets and their evaluation also highlights the advantage of the calibration strategy we proposed and applied here,
which focuses on processes like snow or glaciers rather than on discharge. Calibrated models can compensate for meteorologi-
cal uncertainty and model deficiencies (Refsgaard and Storm, 1996). Including the representation of processes such as snow or

glaciers into the calibration procedure avoids some parameter equifinality and preserves internal relationships (e.g. when SWE
is accurately represented, this should lead to relatively accurate melt contributions to discharge) (e.g. Duethmann et al., 2014;
Finger et al., 2015), which facilitates identifying where models are in need for improvements. Finally, as presented in the Meth-
ods section, the STANDARD and CHELSA meteorological forcing datasets were derived using statistical or physical/heuristic
downscaling, whereas the CERRA-CHELSA dataset is based on a dynamically generated regional reanalysis product. None

of these methods used to create the hyper-resolution forcing datasets led to general improvements in simulation performance
across the full domain. Regional differences between the datasets, observational uncertainty and deficiencies in hydrological
model structure did have a pronounced effect on model performance for discharge.

## 4.2 Applicability and limitations

Our new model setup generally shows a better representation of discharge than the existing hyper-resolution version of PCR-

GLOBWB 2.0 (see Figure 10 D, E, and F). Absolute discharge performance is generally satisfactory, especially at larger spatial
scales and in natural rivers (see Figure 10 A, B, and C) and spatial patterns can be resolved in high detail (see Figure 12 A





**Figure 12.** Illustration of the discharge simulations of the model with examples from hydrologically diverse river stations. (A) Average discharge (1990–2019) over the model domain. (B.) Average discharge (1990–2019) over the Swiss canton of Grisons. The bottom two rows show climatologies (2010–2019) of observed vs. simulated discharge for the glacier-dominated Massa at Blatten (C), the rain-dominated Thur at Halden (D), snow-dominated Inn at Innsbruck (E), and the larger river Danube at Hainburg (F).



and B). Applications of such hyper-resolution LHMs should thus make use of these strengths by studying spatial patterns at larger spatial scales. Still, the model shows poor discharge simulation performance in some catchments, especially in smaller catchments at lower elevations and in heavily regulated rivers. Users should thus be aware of such limitations when looking
at individual catchments. While the hyper-resolution simulations might not be completely accurate in certain places, they can still be valuable in helping to complete the spatial picture and enable studying spatial variability (Seibert et al., 2018b). This is especially true in regions where no observational data are available. There, these LHMs provide a first order estimate of river discharge, without requiring local calibration.

Aside from discharge, LHMs provide also information on other variables such as SWE. Although there exist many local or
national snow reanalysis products that are more accurate than our SWE simulations, semi-global products are often unrepresentative for mountain regions, since their coarse-resolution makes it difficult to represent complex terrain (Mortimer et al., 2020; Mudryk et al., 2024). Our analysis shows that our higher resolution model setup outperforms coarser reanalysis products such as CERRA-Land and ERA5-Land (Figure 5), but does not quite reach the performance of national-scale reanalysis products such as the OSHD product (Figure 5G). Hyper-resolution LHMs could thus provide additional SWE products bridging the
quality gap between national and global datasets.

How suitable is the model for climate impact assessments – a typical application that hydrological models struggle with? While many larger-scale climate change impact assessments still use temperature-index based approaches, including studies focusing on snow (e.g. Kraaijenbrink et al., 2021; Yang et al., 2022) or glaciers (e.g. Kraaijenbrink et al., 2017; Van Tiel et al., 2023; Hanus et al., 2024), there is discussion about the robustness of degree-day approaches under climatic change (Carletti
et al., 2022). Our evaluation, which explicitly addressed model transferability, shows that model performance for discharge and SWE remains mostly consistent over the warmer and colder evaluation periods compared to the reference period (Figure 11). Furthermore, we specifically aimed for increased robustness of parameters by focusing on SWE or glacier melt instead of streamflow during calibration. Sleziak et al. (2020) showed that the stability of hydrological model parameters such as DDFs across periods with different climate conditions can be increased by emphasizing snow representation during calibration. We
therefore have some confidence in the transferability of our model to warmer climate conditions. Still, the general caveats of the model remain applicable when extrapolating into the future. For example, the glacier module shows a realistic dynamic response to climate over larger regions, but not necessarily for each individual glacier (Figure 8). Users should be mindful of this uncertainty in individual glacier responses to climate change. Over time, continued glacier retreat will reduce glacier melt contributions to discharge, since globally most glaciers have either already surpassed their peak in glacier melt contributions or
will do so in the coming decades (Huss and Hock, 2018). This will lead rivers to shift towards more snow-dominated regimes (Farinotti et al., 2012). Further into the future (i.e. near the end of the century) the exact response will thus be less important.

In summary, the presented model setup adds value to regional scale hydrological studies, focusing on general patterns and on larger rivers which are nearly natural to moderately regulated. Caution should be taken when interested in individual, small or heavily regulated rivers, where catchment-scale modelling likely performs better. Still, in regions with limited data availability,
large scale model runs with global parameterization might remain the best option available even for such small rivers.





# 5   Conclusions

Hydrological processes in mountain regions control the supply of water to dependent downstream regions and are therefore important well beyond the fringe of mountains. An accurate representation of these processes in LHMs is thus crucial, especially now that these models are moving towards higher spatial resolutions. In this paper, we proposed different model adjustments to

the well-known and frequently used PCR-GLOBWB 2.0 model in its high-resolution version to make large-scale models fitter for applications in mountain regions. We studied how meteorological forcing and an improved representation of snow, glaciers and soil affect discharge in the larger Alpine region. We conclude that:

- An improved representation of snow and glaciers improves SWE and discharge simulations in high Alpine catchments with natural flow conditions (Hypothesis 1 is supported).

- The introduction of better runoff partitioning in the soil leads to an improvement of discharge simulations in smaller rain-dominated catchments by increasing their flashy response to rainfall peaks. However, these catchments still show overall weak model performance for discharge, suggesting that there is more to gain from improved routing in the soil (Hypothesis 2 is supported).

- Meteorological forcing is uncertain over Alpine regions and different forcing datasets lead to major differences in model
performance for discharge. These differences vary spatially between different forcing datasets.

- Discharge simulations forced by a dynamically-downscaled product (CERRA-CHELSA) did not consistently outperform runs with the other forcing products (STANDARD and CHELSA; Hypothesis 3 is not supported)

- A major control limiting model performance for discharge in Alpine catchments remains water management such as through hydropower plants or water transfers that are not accurately represented in the model, which should be a priority
of future model development.

- Finally, we presented a new model setup with an improved representation of hydrological processes relevant in alpine regions, which is well suited to study regional and larger-scale streamflow and snow patterns in and around mountain regions. This new setup can be used to find answers to remaining questions on water resources, environmental problems and climate impacts in the Alps and around the world.

*Code and data availability.*

**Code**

The existing version of PCR-GLOBWB 2.0 is available on https://github.com/UU-Hydro/PCR-GLOBWB_model. The new model changes will be made available as a separate branch on https://gitlab.ethz.ch/gjanzing/PCR-GLOBWB_model/-/tree/alpine_model?ref_type=heads.



**Data**

We used meteorological data from several different sources. The downscaled STANDARD forcing over the model domain from (Van Jaarsveld et al., 2024) will be uploaded to the repository EnviDat upon acceptance of this manuscript. The source dataset W5E5v2.0 is available at (Lange et al., 2021) and the CHELSA-BIOCLIM+ climatologies are available at Brun et al. (2022a). Forcing from the CHELSA v2.1 can be downloaded from Karger et al. (2021a). This dataset on ENVIDAT will be

updated to cover the full study period. Temperature from CERRA can be downloaded from Schimanke et al. (2021), whereas the precipitation of CERRA-Land can be retrieved from Verrelle et al. (2022). The downscaled CHELSA-CERRA temperature data will be made available on EnviDat upon acceptance of this manuscript. Finally, precipitation from the APGD is available at Isotta and Frei (2013).

The MERIT Hydro DEM is available from Yamazaki et al. (2019). For our model domain, the upscaled version will be

provided on EnviDat upon acceptance of this manuscript together with STANDARD forcing and PCR-GLOBWB 2.0 input. The upscaled MERIT Hydro DEM is also available at (Verkaik and Sutanudjaja, 2024).

Data on discharge, catchment area and reservoirs can be requested from the agencies or downloaded from the sources listed in Table A1. The DEM used for catchment delineation is the COPERNICUS DEM, which is available at European Space Agency and Airbus (2022).

The SWE OSHD dataset for Switzerland is available at Mott (2023) and the SNOWGRID dataset for Austria can be downloaded from GeoSphere Austria (2022). Further snow reanalysis products used are ERA5-Land, available at Copernicus Climate Change Service (2019), and CERRA-Land, available at Verrelle et al. (2022). Finally, estimates of SWE from station data can be downloaded from Fontrodona-Bach et al. (2023a).

Glacier volumes are available from Farinotti (2019). Remotely-sensed glacier elevation changes can be downloaded from

Hugonnet et al. (2021b). Data on mass balances of individual glaciers can be retrieved from World Glacier Monitoring Service (WGMS) (2023) and from GLAMOS - Glacier Monitoring Switzerland (2022). Glacier response times can be downloaded from Zekollari et al. (2020a) and glacier outlines and glacier outlines and areas from GLIMS Consortium (2018). Glacier percentage per catchment can be calculated from the Randolph Glacier Inventory (Pfeffer et al., 2014) and using the catchment outlines in Table A1.

Soil moisture data (v8.1) from the ESACCI can be downloaded from Dorigo et al. (2023).



## Appendix A: Sources for streamflow, catchment shapes and reservoirs

**Table A1.** Agencies and databases as sources for data on streamflow, catchment shapes and reservoirs. 1: DEM used is the COPERNICUS DEM (European Space Agency and Airbus, 2022).

| Country | Data Type | Sources | Link |
|---|---|---|---|
| Austria | Streamflow | Austrian Ministry of Agriculture, Forestry, Regions and Water Management | https://ehyd.gv.at/ |
| | Catchment Shapes | Large-Sample Data for Hydrology and Environmental Sciences for Central Europe (Klingler et al., 2021) | |
| | Reservoirs | Austrian Ministry of Agriculture, Forestry, Regions and Water Management; Simmler (1961); Partl (1977) | https://www.bml.gv.at/ |
| France | Streamflow | Ministry of the Environment, Sustainable Development and Energy (Banque HYDRO) | https://www.hydro.eaufrance.fr/ |
| | Catchment Shapes | delineated from DEM[1] | |
| | Reservoirs | Comité Français des Barrages et Réservoirs | https:www.barrages-cfbr.eu |
| Germany | Streamflow | Bavarian State Office for the Environment and the State Institute for the Environment Baden-Württemberg | https://www.lfu.bayern.de and https://www.lubw.baden-wuerttemberg.de |
| | Catchment Shapes | State Institute for the Environment Baden-Württemberg, delineated from DEM[1] | https://www.lubw.baden-wuerttemberg.de |
| | Reservoirs | Speckhahn et al. (2020, 2021) | |
| Italy | Streamflow | Regional Environmental Agencies from Lombardia, Aosta, and Piemonte | https://www.arpalombardia.it; http://presidi2.regione.vda.it/str_dataview_download and http://www.arpa.piemonte.it |
| | Catchment Shapes | delineated from DEM[1] | |
| | Reservoirs | AQUASTAT Geo-referenced Database on Dams; the Italian Ministry of Infrastructure; OpenStreetMap | https://www.fao.org/aquastat/en/databases/dams; http://dati.mit.gov.it/catalog/dataset/grandi-dighe-italiane |
| Switzerland | Streamflow | Federal Office for the Environment | https://www.bafu.admin.ch |
| | Catchment Shapes | Federal Office for the Environment | https://www.bafu.admin.ch |
| | Reservoirs | Federal Office for the Environment | https://www.bafu.admin.ch |



## Appendix B: Δ h-parameterization

The following detailed description of the Δ h-parameterization is based on Seibert et al. (2018a) and Huss et al. (2010).

Each glacier has a minimum surface elevation $E_{\min}$ and a maximum surface elevation $E_{\max}$. This topographic range of the
glacier surface can be split into $N$ elevation zones (in our case into 20 steps, with $E_{\min}$ and $E_{\max}$ rounded to the nearest multiple
of 50). Each elevation zone $E_i$ with $1 <= i <= N$ is then normalized to :

$$E_{i,\mathrm{norm}} = \frac{E_{\max} - E_i}{E_{\max} - E_{\min}} \tag{B1}$$

The surface elevation of a glacier responds to glacier mass loss in a specific way that depends on its surface elevation. Huss
et al. (2010) provided an empirical relationship between the normalized elevation zone $E_{i,\mathrm{norm}}$ and the normalized (unitless)
change in water equivalent of the glacier ice within this elevation zone $\Delta h_{i,\mathrm{norm}}$;

$$\Delta h_{i,\mathrm{norm}} = (E_{i,\mathrm{norm}} + a)^{\gamma} + b(E_{i,\mathrm{norm}} + a) + c, \tag{B2}$$

with $a$, $b$, $c$, and $\gamma$ as empirical coefficients. Huss et al. (2010) provided three different sets of values for these empirical
coefficients based on the initial surface area of the glacier.

Next, we need to couple this theoretical unitless ice thickness change to the actual observed mass loss. This is done by
means of a scaling factor $f_S$ (m). This scaling factor is the ratio between the total mass loss over the glacier $\Delta M$ (m) and the
integrated normalized change in surface elevation, scaled by the surface area of the elevation zone $A_i$ (as a fraction of the total
glacier area).

$$f_S = \frac{\Delta M}{\sum_{i=1}^{N} A_i * \Delta h_{i,\mathrm{norm}}} \tag{B3}$$

Now, we can compute the new water equivalent for each elevation zone after a certain amount of mass loss.

$$h_{i,k+1} = h_{i,k} + f_S \Delta h_{i,\mathrm{norm}}, \tag{B4}$$

with $k = 0$ as the initial glacier profile and $h_i$ (m) is the ice thickness (m water equivalent) in each cell in that specific elevation
zone. Huss et al. (2010) restricted surface elevation lowering at the edge of the glacier (where $h < 10$ m). Here, we do not
apply this because our grid cells are much coarser than those they used and the ice thickness is thus almost never that thin.
Finally, we make sure that cells can not have less than 0 ice thickness. Any leftover mass loss is distributed over the rest of the
glacier. Note that we apply this procedure to all glaciers independent of size. However, this scheme will hardly affect glaciers
consisting of only a few cells, which are mostly governed by the mass balance per grid cell. For larger glaciers, this scheme
becomes more important.

*Author contributions.* JJ, NW, MvT and MIB contributed to the conceptualization of the study. JJ, BvJ and DNK worked on software and
data curation. JJ performed the formal analyses and visualizations. JJ wrote the original draft, with all authors contributing to reviewing and
editing the final draft. MIB and NW were involved in supervision and MIB acquired the funding for this project.



*Competing interests.* Manuela Brunner and Niko Wanders are editors with HESS. The authors declare no further competing interests.

*Acknowledgements.* We thank the Swiss National Science Foundation (SNSF) for funding this study through project 'Predicting floods and droughts under global change' PZ00P2_201818 (granted to MIB). We want to thank Christoph Schlemper and Jonas Götte for helping with getting access to data. We further thank Tobias Jonas for his valuable suggestions with regard to the snow module.



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
