# Peer review of "Hyper-resolution large-scale hydrological modelling benefits from improved process representation in mountain regions"

_EGUsphere, 2024_

## Author Comment (AC2)

**RC2:** 'Comment on egusphere-2024-3072', **Kristian Förster, 04 Mar 2025**

In their manuscript "Hyper-resolution large-scale hydrological modelling benefits from improved process representation in mountain regions" Joren Janzing et al. describe a set of model extensions to PCR-GLOBWB 2.0 in order to make it more suitable for kilometre-scale modelling in mountainous regions. Together with model improvements in representing snow, glaciers, and soil, different forcing datasets are also tested. In essence, they found that the new model is capable of representing relevant processes in mountain hydrology quite well, even though the original model has been developed for large scale analyses with coarser spatial resolution. The manuscript fits well into the scope of Hydrology and Earth System Sciences. I found that the analyses carried out in the manuscript clearly show the added value of the most recent advances even though I think that some points require some more explanations and / or discussions. Apart from that I believe that the manuscript is an important contribution and I am looking forward to reading the final revised manuscript. Please find my comments below.

**Response:** Thank you for assessing our work and for your careful and constructive feedback. We address your suggestions and concerns below.

General comments

1. Structure of experiments / evaluations. The order of single steps is different throughout the manuscript. Abstract: forcing, snow, glaciers, soil; Introduction: snow and glaciers, soil, forcing. Hypotheses: snow and glaciers, "reviewing standard parameterizations" (I think that H2 should be rephrased in order to better reflect what was actually done. Do you mean reviewing and adjusting?), forcing. Methods and later results: Forcing, snow and glaciers, and soil. Conclusions: Snow and glaciers, soil, forcing. I was wondering if it would help the readers to agree on a consistent order of steps throughout the manuscript?

   **Response:** Thank you for highlighting the need for more consistency in the order of the experiments. We have followed your suggestions and improved consistency by generally following the order "Forcing; Snow; Glaciers; Soil" - this aligns with the order of our experiments. We have changed this in the conclusions and the hypotheses, meaning that the numbers of the hypotheses have changed.

   We make one exception and deviate from this rule in the introduction. The reason for this is that we consider the snow and glacier component to be more important for our storyline than the meteorological forcing.
   We have implemented the following changes in the text to clarify why the specific order was chosen (note that we have also changed the hypothesis regarding the standard parameterization in accordance with your suggestion).

   In section Introduction:
   *"**Nevertheless, any potential gains in model performance due to improved process representation or parameterization are constrained by the quality of the model input. It is thus important that such conditions are met first. H**ydrological modelling*

*is sensitive to the meteorological forcing dataset used as input…"*

*"While large-scale hydrological simulations at higher spatial resolution have become feasible thanks to increasingly available computational resources, it is yet unclear by how much hydrological simulations can improve when combining such high-resolution models **with the latest generation of meteorological forcing datasets and improved process representation**.*
*Therefore, we here aim to explore the effect of **(1)** using different meteorological datasets, **(2)** improving snow and glacier representations, and **(3)** changing runoff partitioning in the soil **on discharge simulations in PCR-GLOBWB 2.0**."*

*"We hypothesize that hyper-resolution LHM performance for discharge in mountain regions will increase by **(H1)** using forcing products that include a representation of smaller-scale atmospheric dynamics compared to other forcing products, **(H2)** improving the representation of mountain hydrological processes, such as snow and ice melt; and **(H3)** reviewing **and adjusting** standard parameterizations."*

In section Conclusion:
*"- Meteorological forcing is uncertain over Alpine regions and different forcing datasets lead to major differences in model performance for discharge. These differences vary spatially between different forcing datasets.*
*- Discharge simulations forced by a reanalysis product using high-resolution atmospheric dynamics at 5.5 km (CERRA-CHELSA) did not consistently outperform runs with the other forcing products which use coarser atmospheric dynamics at 31 km (STANDARD and CHELSA; **Hypothesis 1** is not supported).*
*- An improved representation of snow and glaciers improves SWE and discharge simulations in high Alpine catchments with natural flow conditions (**Hypothesis 2** is supported).*
*- The introduction of better runoff partitioning in the soil leads to an improvement of discharge simulations in smaller rain-dominated catchments by increasing their flashy response to rainfall peaks. However, these catchments still show overall weak model performance for discharge, suggesting that there is more to gain from improved routing in the soil (**Hypothesis 3** is supported)."*

1. Water balance signature in L371 (and also Figure 7): I found the adoption of the Water balance signature not well introduced into the evaluation even though it seems to be a reasonable indicator. Firstly, I would recommend to better explain what positive and negative values mean. From what I could get from the equation (and later by referring to Salwey et al., 2023), it seems to me that a positive value suggests a gain in water, while negative values would indicate a loss of water. Is that true? Please consider adding a short description that could help the readers to better follow your interpretation later. Secondly, I was wondering why you call this water balance signature later in the results section "water gap"? This even more confusing, since positive values seem to be related to additional water rather than a gap. I would suggest to refer to the original term suggested by Salwey et al., 2023.

   **Response:** Thank you for the suggestions. We have changed our terminology to "Water balance signature" to match the terminology used in Salwey et al.,2023 and later in the text mostly refer to it as WB. Furthermore, we have added additional clarification of how

this metric should be interpreted.

Textual changes:

*"**One way** to separate regulated from natural catchments **is the water balance signature (WB), which describes** the deviation from a closed water balance assuming no long-term storage effects (Salwey et al., 2023)."*

*"**Positive values of WB indicate that the catchment discharge is higher than expected. Assuming that the meteorological components could be well-estimated, such positive values could suggest that a catchment gains more water than what comes in through precipitation. Negative values of WB indicate that a catchment loses more water than just the potential evapotranspiration.**"*

*"**However,** other factors such as errors in the meteorological forcing or additional water input from glaciers **due to imbalance** can also lead to strong water balance deviations, which can affect model performance with respect to discharge. **We thus use WB as a general metric to study the effect of such deviations in the water balance on model performance. In addition, we also use WB as an indication for water transfers, hydropower production and other water balance deviations in combination with catchment-based information on reservoirs.**"*

1. Glaciers in Figure 8: I agree that your modelling approach might be more suitable to regions of glaciers rather than individual ones. However, I really went through hard times reading Figure 8. Panels a) and b) are hard to read. Would be worth to increase the zoom level and to make glacier outlines smaller? Where is Mer de Glace (L447) on the map? I would suggest updating Figure 8 a) to d) in order to improve readability. The larger glaciers seem to have a positive mass balance bias, see panels e) and f), while in panel g) the model seem to have quite good average skill for glaciers with an area of more than 8 square kilometers. Why do we see a dramatic depletion in skill when looking at smaller glaciers? Why is 8 square kilometers chosen here? Is it related to the threshold in the Delta h method, proposed by Huss et al. (2010)? I was wondering if a regional mass balance, similar to e) and f), would better support your findings? Moreover, what remains unclear is whether the mismatch in glacier response time is related to differences in model (structure) or related to different forcings, when compared to Zekollari et al. (2020)?

   **Response:** Thank you for making us aware that this. Figure is difficult to understand. We have improved Figure 8 by implementing your helpful suggestions, such as further zooming in on the considered glaciers, showing thin outlines and highlighting some specific glaciers.

   Regarding the additional points you listed:
   - Mass balance bias of larger glaciers: we propose to highlight the Aletsch glacier and Glacier d'Argentiere in plot G and H, to indicate that there is indeed some bias, as you point out. Note however that the mean mass balances are only shown over the Evaluation period II.

   - Depletion of skill for smaller glaciers and for response times compared to Zekollari et al., 2020: Indeed, there is a significant drop in performance for smaller glaciers. We

touched upon the possible reasons for this in the Discussion section.

*"Still, we note that individual and especially smaller glaciers can show significant biases in the mass balance (see Figure 8G) and in the response times (see Figure 8H). The generally shorter response times than in previous experiments might be partly explained by the Δh-parameterization, which ignores potential delays in glacier response to mass changes (Seibert et al., 2018a). Further biases in both mass balance estimates and response times are likely related to the relatively coarse spatial resolution of our model compared to the size of individual glaciers, which makes it more difficult to accurately describe patterns in melt or snow accumulation for small glaciers consisting of only a few grid cells. Melt representation of individual glaciers could be improved by resolving glaciers at higher spatial resolution than the one of the LHM, for example by including elevation zones (Seibert et al., 2018a). Resolving glaciers at higher spatial resolution could also lead to even more realistic glacier retreat, since the Δh-parameterization was originally designed for higher spatial resolutions (Huss et al., 2010) and is likely less accurate at the coarser 30-arcsec model grid."*

- Choice for 8 km2: We had chosen this threshold somewhat arbitrarily to illustrate the effect of glacier size on performance, as we noted a clear relationship between the two. We have now removed this sharp glacier area division and instead show a more continuous relationship between model performance and glacier area to make it less arbitrary.

- Regional mass balance: Thank you for this interesting suggestion. We have considered adding a subplot of a time series of the regional mass balance, similar to 8e and f, but in the end decided against this as this approach would include many additional assumptions. The resulting time series would heavily depend on our selection of glaciers -- of which only a few have (long term) yearly measurements and which might not be equally distributed in space and time.
Instead, we recreate subplot 8g based on geodetic measurements from Hugonnet et al., 2021, which have full spatial coverage and thus allow us to have regional estimates of the mass balance but only over a 10-year time period.

Hugonnet, R., McNabb, R., Berthier, E., Menounos, B., Nuth, C., Girod, L., ... & Kääb, A. (2021). Accelerated global glacier mass loss in the early twenty-first century. *Nature*, *592*(7856), 726-731.

1. Glacier initialization in 1990. How do initialize glaciers (area, volume) in your model setup when you deviate from the calibration period (which coincides with the data in Farinotti et al., 2019 for the early 2000)?

   **Response:** We indeed use the consensus estimate from Farinotti et al., 2019, which are representative for the early 2000s, to initialize our glaciers in 1990. We agree that this may lead to inconsistencies in glacier outlines around 2000. Therefore, we changed the implementation and keep the glacier volume and outline constant until 2000, although we still calculate ice melt and accumulation. This new approach is largely similar to previous large-scale hydrological modelling studies (e.g. Hanus et al., 2024).
   We acknowledge that even this updated procedure is still not optimal. However, to our

knowledge, there are no glacier volumes/ice thickness available representing conditions before 1990 and are available globally or continentally (needed to easily apply the model to other regions).

We have added some further clarification of the new approach in the text:

In section Datasets:
"*As we implement a new glacier module, we had to define the locations of the glaciers and their initial thickness. This information was derived from the consensus estimates from Farinotti et al., 2019, which are representative for the year 2003 for most glaciers. To our knowledge, there are no estimates of glacier volumes available that go further back in time and that cover all glaciers on the global, continental or Alpine scale. This glacier volume dataset has previously also been used to initialize glaciers for the same study period by Hanus et al., 2024.*"

In section Glacier module
"*To account for such temporal changes, we implement the empirical Δh-parameterization scheme from Huss et al., 2010, as implemented in HBV (Seibert et al., 2018). As the glacier volumes are representative for around the year 2003, we only apply this scheme after the year 2000, keeping the glacier volume and area constant between 1990-2000 while still calculating the ice melt and accumulation.*"

Farinotti, D., Huss, M., Fürst, J.J., Landmann, J., Machguth, H., Maussion, F., Pandit, A., 2019. A consensus estimate for the ice thickness distribution of all glaciers on Earth. Nature Geoscience. https://doi.org/10.1038/s41561-019-0300-3

Hanus, S., Schuster, L., Burek, P., Maussion, F., Wada, Y., & Viviroli, D. (2024). Coupling a large-scale glacier and hydrological model (OGGM v1. 5.3 and CWatM V1. 08)–Towards an improved representation of mountain water resources in global assessments. *Geoscientific Model Development*, *17*(13), 5123-5144.

Specific comments:

L15: What do you mean by unidirectional?
**Response:** We clarified that we mean that changes can be both positive and negative:

"*Our evaluation of the effect of these different adjustments on model performance for discharge shows that while the meteorological forcing has a major effect on discharge simulations, it results in a mixed pattern of performance gains and losses over the domain.*"

L28: "thanks to heatwave". In my opinion you should highlight that this was only possible through a very negative mass balance.
**Response:** We have addressed this point by highlighting in the text that the mass balances were very negative.
"*...whereas Alpine glaciers provided surrounding rivers with surplus melt-water during the 2003 Central European Drought due to very negative mass balances (Van Tiel et al., 2023).*"

L60pp.: I think that there are lots of other references that support your statement here.
**Response:** Thank you for pointing out the need to include additional literature for this statement. We have added several references to support this statement, and have also specified that with "a constant DDF" we refer to a "time-constant DDF".

"However, LHMs often use very simplistic temperature-index schemes e.g. by using only **a time-constant** DDF or by omitting calibration **(e.g. Gosling and Arnell, 2011; Sutanudjaja et al., 2018; Müller-Schmied et al., 2021; Stacke and Hagemann, 2021)**."

Gosling, S. N., & Arnell, N. W. (2011). Simulating current global river runoff with a global hydrological model: model revisions, validation, and sensitivity analysis. Hydrological Processes, 25(7), 1129-1145.
Müller Schmied, H., Cáceres, D., Eisner, S., Flörke, M., Herbert, C., Niemann, C., ... & Döll, P. (2021). The global water resources and use model WaterGAP v2. 2d: Model description and evaluation. Geoscientific Model Development, 14(2), 1037-1079.
Stacke, T., & Hagemann, S. (2021). HydroPy (v1. 0): A new global hydrology model written in Python. Geoscientific Model Development, 14(12), 7795-7816.
Sutanudjaja, E. H., Van Beek, R., Wanders, N., Wada, Y., Bosmans, J. H., Drost, N., ... & Bierkens, M. F. (2018). PCR-GLOBWB 2: a 5 arcmin global hydrological and water resources model. Geoscientific Model Development, 11(6), 2429-2453.

L74pp.: Please explain where exactly theses percentage values refer to.
**Response:** We have further clarified to which gauging stations these percentages refer to:

*"Glaciers can be an important additional water source during their melt season (e.g. the average glacier storage change contribution to total runoff near the river mouth for the rivers Rhone **at Beaucaire** in August: 25%, Rhine **at Lobith** in August: 7%, Danube **at Ceatal Izmail** in September: 4%, Huss, 2011)."*

L77pp.: Introducing one-way coupling would also suggest that two-way-coupling exists, too (see Pesci et al., 2023).
**Response:** We have removed the specification "one-way" to acknowledge this, as the general set-up (using glacier output into the model) is still the same.

*"b.) using the output from an external glacier model as the input to the hydrological model (i.e.*  *coupling the models; "coupled models")."*

L113. I felt some statement is missing here before introducing meteorological forcing.
**Response:** Thank you for your suggestion. We have added a sentence to smoothen the transition from the previous paragraph to the one on meteorological forcing:

*"**Nevertheless, any potential gains in model performance due to improved process representation or parameterization are constrained by the quality of the model input. It is thus important that such conditions are met first.** Hydrological modelling is **indeed** sensitive to..."*

L135: Please consider rephrasing H2 (as suggested before).
**Response:** Following your earlier comment, we have changed H2 in the way you suggested (it is H3 now):
*"...**(H3)** reviewing **and adjusting** standard parameterizations;..."*

L176: I am not sure whether "to guarantee" is correct in this context.
**Response:** Thank you for highlighting this. We have changed the word "guarantee" to "generate more":

*"In contrast, near-surface air temperature can be further downscaled to **generate more** accurate spatial melt patterns at 30 arcsec resolution."*

L239: DDF in general depend on land use. For the large-scale model it's clear that working with an "average" DDF makes sense. However, I remember from own research that DDF shows a quite different behavior in forests. Given the increased resolution and the relevance of forests in the water balance in mountainous areas, I think it is at least worth to discuss this later.
**Response:** We have expanded our discussion on spatial patterns in DDFs by explicitly discussing a potential role for land cover and vegetation.

In section Discussion:
*"Another reason for this slight decrease in discharge performance at lower elevations could be related to our choice of regionally-averaged DDFs, since in reality these DDFs show smaller-scale variability in space (e.g. with aspect, albedo, elevation, **landcover, vegetation,**...) **(Kuusisto, 1980; Rango and Martinec, 1995;** Hock, 2003; Ismail et al., 2023)"*

*"More elaborate snow module formulations, such as parameterizations that include aspect (e.g. Immerzeel et al., 2012) or radiation (e.g. Hock, 1999), could increase our ability to capture more detailed spatial melt patterns. **Such approaches can become feasible now that higher spatial resolutions can resolve slopes and represent vegetation cover and land use in a more detailed way.** However, they come at the cost of increased model complexity and a larger number of input variables."*

Kuusisto, E. (1980). On the values and variability of degree-day melting factor in Finland. Hydrology Research, 11(5), 235-242.
Rango, A., & Martinec, J. (1995). Revisiting the degree-day method for snowmelt computations 1. JAWRA Journal of the American Water Resources Association, 31(4), 657-669.

L261p: I do not understand that snow is transported to glaciers, given that transport is only considered to non-glaciated cells. Could you please explain this please?
**Response:** Thank you for highlighting the need for further clarification. Based on this comment and comments from the other reviewers, we have added further details:

*"This means that snow can **only** be transported **from a.) a non-glacierized cell to a non-glacierized cell and b.) from a non-glacierized cell to a glacierized cell. There is no snow transport from a glacierized cell to either a glacierized cell or a non-glacierized cell. When snow is transported onto a glacierized cell, it is added to the snow cover and can thus later become part of glacier accumulation (Kuhn, 2003, Freudiger et al., 2017).**"*

Freudiger, D., Kohn, I., Seibert, J., Stahl, K., & Weiler, M. (2017). Snow redistribution for the hydrological modeling of alpine catchments. *Wiley Interdisciplinary Reviews: Water*, *4*(5), e1232.
Kuhn, M. (2003). Redistribution of snow and glacier mass balance from a hydrometeorological model. Journal of Hydrology, 282(1-4), 95-103.

L271: Do you mean lower albedo?
**Response:** Thank you for making us aware of this inconsistency. Indeed, we meant lower albedo and have changed this in the text.

L294: If glaciers can only grow to its original extent, how do you realize to initialize the glaciers with a larger extent in the past?
**Response:** As pointed out above, this is a severe limitation caused by the lack of glacier volumes available further back in time. We thus keep the glacier volume constant until 2000, when we assume the volumes are the same as in the observed datasets.
We refer to the same added text as mentioned above:

*"This information was derived from the consensus estimates from Farinotti et al., 2019, which are representative for the year 2003 for most glaciers. To our knowledge, there are no estimates of glacier volumes available that go further back in time and that cover all glaciers on the global, continental or Alpine scale. This glacier volume dataset has previously also been used to initialize glaciers for the same study period by Hanus et al., 2024."*

L311: I am sorry if I missed this important detail but for me it's not clear if you calibrate a single parameter (e.g., DDFmin) for the entire domain or catchment-wise. From the table in supplement, it seems to me that it's a single value only which is valid for the entire domain after calibration.
**Response:** Thank you for raising this point. Indeed, we calibrate these parameters for the domain as a whole. We added some additional clarification.

*"We calibrate on the specific process (e.g. on SWE) instead of discharge to avoid compensating with parameter calibration for deficiencies in other processes and to increase the stability of these parameters under varying temperatures (Sleziak et al., 2020). We try to find one parameter set that is regionally valid, **i.e. constant over the entire domain**."*

L378: Why is snow calibrated only in Switzerland (I hope that I understood it correctly)? Here you focus on both Switzerland and Austria?
**Response:** Thank you for raising this point. We will further clarify that we do this because we are interested in testing the transferability of the model to other regions, as we do not have comparable snow products for other parts of the Alps such as Italy or France.

In section Calibration:
*"… and because Switzerland covers diverse climatic regions. **Note that we do not calibrate over Austria, even though we have similar data available there as well, to be able to test the transferability of the snow scheme to other regions.** To explicitly account for elevation-dependent melt patterns,…"*

In section Evaluation:

*"In addition to discharge, we evaluate model performance for SWE by calculating spatial averages over different elevation bands (0-1000 m, 1000-2000 m, 2000-3000 m or the entire country) over Switzerland and **also** over Austria, **as reference for domains where the model is not calibrated.**"*

L383: How do you evaluate glacier geometry evolution? Later you explain that this was only done visually.

**Response:** Thank you for raising this point. We have highlighted that we evaluate the glacier geometry evolution both visually to check the plausibility of spatial patterns and over longer time intervals compared to Zekollari et al., 2020**.**

*"To evaluate glacier **geometry evolution,** we both visually compare spatial patterns of glacier surface elevation changes against observations (GECM; see Table 2) **and quantitatively evaluate their long-term changes**. It is difficult to evaluate the long-term response of simulated glaciers to climate forcing given the relatively short study period."*

Zekollari, H., Huss, M., Farinotti, D., 2020. On the Imbalance and Response Time of Glaciers in the European Alps. Geophysical Research Letters 47, e2019GL085578. https://doi.org/10.1029/2019GL085578

L442p: Her, I think it would be interesting to provide a few more details what we see, e.g., the values around zero for small snow fractions. By the way: Is snow fraction the percentage of precipitation that is solid or how is it defined?

**Response:** Based on your comment and comments from other reviewers, we have provided some further details about the results and also indicated in the text that snow fraction indeed refers to the percentage of precipitation that is solid. We have further clarified this by using the term "snowfall fraction" and adding an explanation to the text.

*"Figure 7A, B, and C illustrate that the **changes in the snow module** mainly **improve** model performance for discharge in **catchments with high snowfall fractions, whereas in catchments with low snowfall fractions the changes are negligible or slightly negative. Still, some catchments with high snowfall fractions experience decreases in performance for discharge: these decreases mostly happens in the presence of reservoirs and/or negative values of WB.**"*

L491pp: When comparing the drop in median for the different periods, it would be relevant to mention them in the text. Given the very good representation of SWE in the warmer period, is there any observation regarding the glaciers?

**Response:** Thank you for your comment. Based on comments from Reviewer 1, we have decided to place less focus on this section and moved the Figure to the Appendix. We add a subplot on comparing glacier performance between the different periods. We also add a quantification of the changes to the text in the Supporting Information.

Figure 12: Adding KGE or some other score would be helpful. Why does the full model compute a very high peak for Massa. Given that it is a 10 yrs. average, it seems to be a quite high event.

**Response:** We have added a KGE score for the benchmark and the final run to the figure.

We have further investigated the peak for the Massa river and it appears to be related to the choice that we move all the snow on the same day (i.e. beginning of the hydrological year), which led to increased melt around that day. We have rerun the runs with the glacier module, now remaining closer to the original method by Huss et al., 2010 and remove the snow continuously, which also removes this peak in discharge.

Huss, M., Jouvet, G., Farinotti, D., Bauder, A., 2010. Future high-mountain hydrology: a new parameterization of glacier retreat. Hydrology and Earth System Sciences 14, 815–829.

https://doi.org/10.5194/hess-14-815-2010

Technical comments:

Figure 4: Unfortunately, the legend hides important details of the distribution. Would it be possible to make its background transparent? See also Figure 5, 6, 11
**Response:** Thank you for the suggestion. We have moved the legend outside of the distributions plot for all suggested figures.

Figure 10: KGE refers to the upper row A-C, while KGESS to the bottom row (D-F)?

 **Response:**  That is indeed correct. We have further highlighted this in the Figure caption.
*"Model performance of the Full model run for discharge (top row**, showing KGE**) and the total model performance changes for discharge compared to the CERRA-CHELSA benchmark run (bottom row**, showing KGESS**) in the evaluation catchments in relation to different catchment characteristics."*

References

Farinotti, D., Huss, M., Fürst, J.J., Landmann, J., Machguth, H., Maussion, F., Pandit, A., 2019. A consensus estimate for the ice thickness distribution of all glaciers on Earth. Nature Geoscience. https://doi.org/10.1038/s41561-019-0300-3

Huss, M., Jouvet, G., Farinotti, D., Bauder, A., 2010. Future high-mountain hydrology: a new parameterization of glacier retreat. Hydrology and Earth System Sciences 14, 815–829. https://doi.org/10.5194/hess-14-815-2010

Pesci, M.H., Overberg, P.S., Bosshard, T., Förster, K., 2023. From global glacier modeling to catchment hydrology: bridging the gap with the WaSiM-OGGM coupling scheme. Frontiers in Water 5, 1296344. https://doi.org/10.3389/frwa.2023.1296344

Salwey, S., Coxon, G., Pianosi, F., Singer, M.B., Hutton, C., 2023. National-Scale Detection of Reservoir Impacts Through Hydrological Signatures. Water Resources Research 59, e2022WR033893. https://doi.org/10.1029/2022WR033893

Zekollari, H., Huss, M., Farinotti, D., 2020. On the Imbalance and Response Time of Glaciers in the European Alps. Geophysical Research Letters 47, e2019GL085578.
https://doi.org/10.1029/2019GL085578

---

## Author Response (AR1)

Dear Dr. Hendricks Franssen,

Thank you very much for taking the time to reassess our updated manuscript. We very much appreciate the clear and constructive feedback by the reviewers and editor.

We have further supported our approach and conclusions in the following major ways:
i) We have better organized the structure of the document, in particular by maintaining more consistency in the order of that we present our methods and results, by moving the section on transferability to the Supporting Information, and by organizing the used datasets into the categories of input, evaluation and ancillary data, ii) we further explained why we used the Alpine region as an example of a region that is often implicitly modelled by global hydrological models, iii) we elaborated on how we set-up and use the glaciers in the model and made some changes to the way they were implemented, and iv) we further established a link between the water balance signature and regulation, explained better which other processes might influence this and have been more explicit in the way we derive our conclusions from this.

Finally, we wanted to point out that we re-executed some of the model simulations. This can lead to some differences in the results compared to the first draft. We also noticed there were a few gauges falling outside of our chosen region (e.g. the upstream part of the catchments of the Danube, Rhone, Rhine and Po), which is why we reduced the number of references gauges.

Thanks again for reconsidering our manuscript. Below we will respond to the reviewer comments in more detail.

Sincerely,

Joren Janzing
on behalf of the authors

----------------------- **Review 1** -------------------------------------------------------------------------------------
**RC1:** 'Comment on egusphere-2024-3072', **Anonymous Referee #1, 17 Feb 2025**

**Main comment**

The authors take a global hydrological model (PCR-GLOBWB 2.0) and apply it over a large Alpine region. They propose several model adjustments to improve the model in order to better capture snow cover and discharge dynamics. I find it a very interesting study on a relevant topic and I see great potential in their work, however, I do not think that the analysis and manuscript at its current state fulfills the journal's requirements and some clarifications and revisions are required.

**Response:** Thank you for your careful assessment of our manuscript, we very much appreciate your detailed and constructive feedback. Below we will address your suggested points and clarifications one by one.

As a general point, we wanted to point out that we re-executed some of the model runs, which led to some slightly different results than in the first draft. We also noticed there were a few gauges falling outside of our chosen region (e.g. the upstream part of the catchments of the Danube, Rhone, Rhine and Po), which is why we reduced the number of references gauges.

In my opinion, the strength of the manuscript (i.e. the usage of a large selection of forcing and evaluation data sets and the investigation of several processes) is its weakness at the same time, as it is getting more and more difficult to keep a concise and understandable workflow.

**Response:** Thank you for pointing out the need for an understandable workflow. We agree that our manuscript would benefit from a clearer structure and increased conciseness. We have addressed this by removing superfluous steps and increasing the consistency and readability of our text:

First, we moved the model transferability analysis, including Figure 11, to the Supporting Information, because it was not the main focus of the paper.
Second, we have restructured the "Forcing and datasets" section. We have now called the full section "Datasets", which we subdivided into subsections related to "Model input", "Evaluation and calibration data" or "Ancillary data". These sections will help the reader to understand why we used each dataset.
Third, we made sure to follow the same order of the steps in each section as in the experiments, that is "Forcing; Snow; Glaciers; Soil". The increased consistency will help readers to better follow the workflow as described in the manuscript. Accordingly, we have changed the order and numbers of the hypotheses and the conclusions. The one exception is the introduction, as we think that the main relevance is in the snow and glacier components and that these should therefore be mentioned first. We highlight this in the introduction:

*"Nevertheless, any potential gains in model performance due to improved process representation or parameterization are constrained by the quality of the model input. It is thus important that such conditions are met first." (Line 116-117)*

Furthermore, I lack the overall justification of applying a global hydrological model to the Alpine regional scale, as there are better suited models available for this task. Why not using models that have the required process implemented already?

**Response:** Thank you for pointing out that the model choice requires further justification. Global hydrological models are run over larger spatial domains, which also include mountain regions. However, these regions are often not well represented in such large-scale models.

First, we want to test such a global-scale model over a mountain region which is usually implicitly modelled in larger (global) scale applications. We decided to use the Alpine region as an example of such a mountain region and use the domain to evaluate the effects and limitations of different model adjustments, which can in principle be applied at larger scales.

Second, we want to highlight that other large-scale modelling approaches also have difficulty in representing mountain regions. One of the reasons for this is that the smaller Alpine catchments with large elevation differences generally require higher resolution models with more advanced structures (Gurtz et al., 2003), so that detailed (sub-)national scale modelling approaches remain extremely popular in Alpine countries like Switzerland (Horton et al., 2022). Unfortunately, such spatial fragmentation precludes Alpine-wide studies, which is why improved kilometer-scale resolution larger-scale modelling approaches – as the one presented in this manuscript - can facilitate Alpine-scale spatial comparisons.

We have added the following statement to the introduction:

*"We focus on the larger Alpine region, as an example of a mountain region that is normally implicitly simulated by LHMs. A similar set-up can in principle be applied at larger scales. While more detailed hydrological modelling approaches are available for the Alps at the national or catchment scale, the Alps are very rarely studied as a whole at a similarly high spatial resolution, which inhibits comparisons between different regions within the Alps." (Line 137-141)*

Gurtz, J., Zappa, M., Jasper, K., Lang, H., Verbunt, M., Badoux, A., & Vitvar, T. (2003). A comparative study in modelling runoff and its components in two mountainous catchments. *Hydrological Processes, 17*(2), 297-311.

Horton, P., Schaefli, B., & Kauzlaric, M. (2022). Why do we have so many different hydrological models? A review based on the case of Switzerland. *Wiley Interdisciplinary Reviews: Water, 9*(1), e1574.

**Specific comments**

Title: Why is it hyper-resolution? If if would be a global application, then a 1km resolution run is termed hyper-resolution, I think. However, I do not understand why a regional application o the Alps in 1km should be termed hyper-resolution. Please clarify. Please be aware that in the snow-hydrological community, snow simulations in 1km are considered very course and not adequate to capture snow processes.

**Response:** Thank you for highlighting the need for further clarification. Indeed, within the global or continental scale modelling community, a 1km resolution is considered to be hyper-resolution. As mentioned earlier, we framed our study in the context of such large-scale hydrological models, with the Alpine domain serving as an example region, where high spatial resolution could provide added skill as it enables capturing the complex topography. A similar hyper-resolution model set-up can be easily expanded towards the European or global domain.

We are aware that a 1km resolution can be considered very coarse from the perspective of the snow hydrological community. However, high-resolution models applied in this community are often developed for local rather than larger scale applications. Since a trade-off exists between spatial domain and resolution, snow hydrological products at larger scales are also often relatively coarse. For example, Kraaijenbrink et al. (2021) modelled snow and snowmelt over the Himalaya at 5.7 km resolution and Mortimer et al. (2020) mention that the typical resolution of large to global scale SWE products is 25 to 100 km. To clarify the focus of our study (i.e. large-scale applications) we therefore refer to "hyper-resolution large-scale" hydrological modelling in the title to clarify that the focus is not on local-scale "hyper-resolution" modelling performed over a limited domain.

We have now further clarified this point in the introduction (same as the previous point):

*"We focus on the larger Alpine region, as an example of a mountain region that is normally implicitly simulated by LHMs. **A similar set-up can in principle be applied at larger scales.** While more detailed hydrological modelling approaches are available for the Alps at the national or catchment scale, t**he Alps are very rarely studied as a whole at a similarly high spatial***

***resolution,** which inhibits comparisons between different regions within the Alps." (Line 137-141)*

Kraaijenbrink, P. D., Stigter, E. E., Yao, T., & Immerzeel, W. W. (2021). Climate change decisive for Asia's snow meltwater supply. *Nature Climate Change*, *11*(7), 591-597.

Mortimer, C., Mudryk, L., Derksen, C., Luojus, K., Brown, R., Kelly, R., & Tedesco, M. (2020). Evaluation of long-term Northern Hemisphere snow water equivalent products. *The Cryosphere*, *14*(5), 1579-1594.

Line 1: I was wondering. Are there actually major rivers that do not originate in mountain regions?

**Response:** Thank you for raising this question. There are indeed major rivers not originating in mountain regions. To provide just one example, the Volga river in Russia originates at an elevation of just 228 m (Litvinov et al., 2009). This river  encounters some tributaries further downstream that source from the Ural mountains.

Litvinov, A. S., Mineeva, N. M., Papchenkov, V. G., Korneva, L. G., Lazareva, V., Shcherbina, G. K., & Shurganova, G. V. (2009). Volga river basin. Rivers of Europe, 2, 1.

Line 8: Please add in the abstract what is hyper-resolution for you? Please provide numbers.

**Response:** Thank you for pointing this out. We have now specified that the model runs at a 30 arc-sec resolution (approx. 1km).

Line 140: I do not understand 'calibrating SWE agains a reagional SWE'. Usually one calibrates parameters such as the DDF using measured snow. Please clarify.

**Response:** Thank you for your comment. As we are working over a large domain with grid cells of 1 by 1km, we decided to calibrate against a regional SWE reanalysis product that assimilates underlying observations and has both a high resolution and high quality.  Alternatively, we could have calibrated against point measurements or satellite data. However,  point measurements at stations are not representative at the spatial scale of the modelling effort and satellite observations of snow cover fraction might not contain information on the quantity of water available in mountain regions. We clarified the statement in the introduction:
*"…by calibrating SWE against a **detailed** regional SWE reanalysis product **with assimilated observations**…" (L147-148)*

Fig. 1: For me this map was a bit misleading, as I was expecting you to simulate the runoff of the large rivers, but as I understood, you do not look at discharge from those rivers. This map shows another area than what you actually analyze and simulate. Please consider to adapt and show the exact modelling domain.

**Response:** Thank you for raising this point. The way our model is set up, we do simulate the entire domain depicted in Figure 1 and we have grid cells representing the whole domain. Our analysis is indeed mostly focused on the comparison at different sets of observational stations, with some stations located on the rivers depicted. However, soil moisture is evaluated over the entire domain in Figure 9 and we also depict model results for the full domain in Figure 12.

We have updated Figure 1 to show both the model domain and added subplots with the locations of the stations where the model is evaluated.

Fig. 2: As far as I understand, the implementation of the snow transport was not part of this study and hence should be part of the benchmark model in my opinion. The presentation and evaluation of the snow transport scheme was done in another study, right? The removal of this step of complexity in the analysis also could make your total analytical set-up more concise.

**Response:** Thank you for your suggestion. The snow transport implementation was indeed introduced in Van Jaarsveld et al., 2025. However, that paper evaluated snow performance against snow cover fraction only. Here, we explicitly evaluate the added value of this specific step using SWE data. This enables us to quantify its effect relative to the effect of the other development steps and highlights the benefit of the snow transport step in terms of removing the snow tower problem (i.e. continuous build up of higher and higher SWE values over time). We think that it is important to show how snow transport influences the presence of snow towers, as many models do struggle with such unrealistic features. Therefore, we retained this step as a separate analysis step.

Van Jaarsveld, B., Wanders, N., Sutanudjaja, E. H., Hoch, J., Droppers, B., Janzing, J., ... & Bierkens, M. F. (2025). A first attempt to model global hydrology at hyper-resolution. *Earth System Dynamics*, *16*(1), 29-54.

Line 259: What is the SWE threshold used in this study? Please state and explain how it was derived.

**Response:** Thank you for highlighting the need for clarification. We used a threshold of 0.625 m as in Van Jaarsveld et al., 2025, which is based on Frey and Holzmann, 2015. We have specified this in the text:

*"This scheme transports part of the snow downhill based on the surface slope whenever the snowcover **exceeds a SWE content of 0.625m. This threshold is based on values for the forest snow holding capacity and snow density in Frey and Holzmann (2015), using a similar approach as in CWatM (Burek et al., 2020).**" (L277-279)*

Frey, S., & Holzmann, H. (2015). A conceptual, distributed snow redistribution model. *Hydrology and Earth System Sciences*, *19*(11), 4517-4530.
Van Jaarsveld, B., Wanders, N., Sutanudjaja, E. H., Hoch, J., Droppers, B., Janzing, J., ... & Bierkens, M. F. (2025). A first attempt to model global hydrology at hyper-resolution. *Earth System Dynamics*, *16*(1), 29-54.

Line 310: You add the snow routing and calibrate it offline. What were the other model parameters of the model calibrated on? If they were calibrated on discharge, do not all parameters have to be re-calibrated again?

**Response:** Thank you for your comment. The original setup of PCR-GLOBWB has not been calibrated (Sutanudjaja et al., 2018) and the other parameters were derived using information on land cover and other types of spatial information. For example, the soil parameters are derived from existing landcover maps (Hoch et al., 2023). We clarified this in the text as follows:

*"PCR-GLOBWB 2.0 **has not been** calibrated **and generally uses parameters derived from external datasets (Sutanudjaja et al., 2018). Still,** we do calibrate the degree-day factors of snow and ice to increase regional applicability and to ensure realistic glacier geometry evolution." (L335-337)*

Hoch, J. M., Sutanudjaja, E. H., Wanders, N., Van Beek, R. L., & Bierkens, M. F. (2023). Hyper-resolution PCR-GLOBWB: opportunities and challenges from refining model spatial resolution to 1 km over the European continent. *Hydrology and Earth System Sciences*, *27*(6), 1383-1401.

Sutanudjaja, E. H., Van Beek, R., Wanders, N., Wada, Y., Bosmans, J. H., Drost, N., ... & Bierkens, M. F. (2018). PCR-GLOBWB 2: a 5 arcmin global hydrological and water resources model. *Geoscientific Model Development*, *11*(6), 2429-2453.

Line 313: Please be aware that the SWE data products you use also only are model output and the also these models are (for different reasons) often incorrect.

**Response:** Thank you for pointing this out. We acknowledge that the SWE reanalysis product is still a modelled product. As mentioned before, we used this data product because it matched the purpose of our analysis best and because sufficiently detailed observational products over the Alps do not exist. We worked with this dataset, because it has high quality, assimilates underlying observations, and has regional spatial coverage.

We justified this choice in the text as follows:

*"...we used a snow water equivalent reanalysis product over Switzerland (Mott et al., 2023). We chose this dataset because of its **extensive data assimilation,** its spatial continuity, its high quality which is unmet by products covering larger spatial domains (e.g. ERA5-Land or CERRA-Land), and because Switzerland covers diverse climatic regions. **However, as every other SWE dataset, this product also comes with some uncertainties."** (L347-350)*

Line 342: What different climatic regimes are meant here?
**Response:** We will further clarify this and specify that we tested for the transferability of the model to time periods with a higher or a lower average temperature:
*"...we use both evaluation periods to assess the transferability of the new model set up to **time periods with different mean temperatures** within the framework..." (L369-371)*

As mentioned under the general comments, we will move this part of the analysis to the appendix and only briefly mention it in the text in order to reduce complexity.

Line 370: I am a bit skeptical about the applicability of the WB measure in the Alps to assess the influence of reservoirs. As you mention, it is also strongly impacted, e.g., by the the meteorological data used. In my opinion, your results (e.g. Fig. 7) showing the general deviations of the WB from zero are more an indication of the big uncertainties in precipitation and evapotranspoiration. Hence, the WB is a poor measure for reservoir influence and I am not sure what is then the validity of this measure to stay in the manuscript. To me the calculation of the WB does not provide new insights and only adds an unnecessary level of complexity to the study. Please think again what is the added value of calculating and showing the WB so prominently.

**Response:** Thank you for raising this point and asking us to reflect on the added value of this

metric. The WB signature is indeed influenced by other processes than regulation such as uncertainties related to meteorological forcing.

Still, we argue that there is value in using this metric in our manuscript:
1. The WB signature is related to the presence of strong regulation as shown in Figure i in this response. The hidden variable that confuses the picture might be the "degree of regulation" or the "type of regulation". There are many catchments with reservoirs that are not so heavily regulated, which can obscure some of the results. However, Figure i, which we added to the Supporting Information, clearly demonstrates that the degree of regulation (here defined as "the total reservoir volume over the catchment divided by the mean annual discharge"; this results in the amount of time the reservoirs can store the catchment discharge) is generally related to negative values of the WB signature and on average to stronger deviations from 0.
However, this finding clearly does not hold for all catchments, as not all catchments experience strong water abstractions or inter-catchment water transfers. Salwey et al., (2023) – who introduced the metric – acknowledge that there is no one-size-fits-all indicator of reservoir regulation on streamflow. Therefore, we think that the presence of reservoirs and the WB signature can together provide complementary information on the effect of regulation on model performance.

[Figure]

*Figure i Relationship between the degree of reservoir regulation and the WB signature. The degree of reservoir regulation is here defined as the total reservoir volume over the catchment divided by the mean annual discharge.*

2. There is a clear dependence of model performance changes on this WB metric (see Figure 7 and 10 in the manuscript). Performance mostly increases when this metric is positive and decreases when this metric is negative. Regardless of the relationship to reservoir regulation, this adds value by itself.
Deviations from 0 in the metric for unregalated basins can indicate that also other influences - such as biases in meteorological forcing – can significantly affect model performance and that our model can often not compensate in catchments where such biases exist. Furthermore, the metric can identify catchments with a significant meltwater input related to glacier mass loss.

We do acknowledge that the choice of this metric has to be better justified in the manuscript and propose the following modifications and clarifications.

1. We redefine the term "regulated catchment" in the revised version of the manuscript. Instead of just showing if a reservoir is present, we now define a threshold based on the degree of regulation using Figure i. Catchments are classified as regulated when their degree of regulation exceeds 0.1 years (threshold based on the first change of the median in Figure i): this change will avoid classifying catchments with little regulation as regulated.

2. We improve the separation of conclusions derived from the analyses using the WB signature only and conclusions for regulated catchments derived from analyses jointly considering the WB signature and information on reservoirs. We also added quantification for regulated and unregulated catchments in the text. We further highlight in the text that WB anomalies can for example be related to biases in meteorological forcing and glacier mass loss.

Textual changes:

"*However, other factors such as errors in the meteorological forcing or additional water input from glaciers **due to imbalance** can also lead to strong water balance deviations**, which can also affect model performance with respect to discharge. We thus use WB as a general metric to study the effect of such deviations in the water balance on model performance. In addition, we also use WB as an indication for water transfers, hydropower production and other water balance deviations in combination with catchment-based information on reservoirs.**"* (L411-415)

"*A major control limiting model performance for discharge in Alpine catchments **are strong deviations from a closed water balance. Such deviations are sometimes related to** hydropower plants or water transfers that are not accurately represented in the current model setup, which should be a priority of future model development.*" (L704-706)

Salwey, S., Coxon, G., Pianosi, F., Singer, M. B., & Hutton, C. (2023). National-scale detection of reservoir impacts through hydrological signatures. *Water Resources Research*, *59*(5), e2022WR033893.

Line 402: In my opinion, the comparison of the meteorological input data sets with regard to the discharge performance (Fig. 4 G,H and I) should be conducted after the model routines have been improved. As seen in Fig. 4, the overall model performance seems fairly low with a median KGE barely above 0. If the models routine is not good enough, also a better precipitation input, for example, cannot improve runoff in a snowmelt-dominated catchment. Please think of moving the evaluation of the different meteorological input data sets at the end of you workflow.

**Response:** Thank you for your suggestion. We appreciate the feedback and have carefully considered moving the meteorological analysis to the end of the analysis. However, we have finally decided to keep the order of experiments as in the original manuscript because of the following reasons:

First, the main part of the model does not change by changing the input dataset: we do not alter any parameters related to landcover, groundwater, routing or human water use. As mentioned before, PCR-GLOBWB is generally not calibrated (Sutanudjaja et al., 2018) and these parameters are derived from specific data sources, e.g. from datasets on landcover. Second, the largest changes in model performance we found were related to meteorological products, albeit with mixed performance changes. Although improving the different model routines leads to general model performance gains, the model forcing has a major effect on

model performance. This highlights that a model can only improve as much as the quality of the model input allows. Therefore, we think that it is crucial to first identify the best forcing dataset. Once this dataset has been identified, we then test various other ways of potential model improvement by adjusting different model routines. Furthermore, as the forcing dataset has a very large effect compared to the other modelling choices, the general differences in model performance between meteorological forcing datasets would persist regardless of when we evaluate the various meteorological forcings.

Sutanudjaja, E. H., Van Beek, R., Wanders, N., Wada, Y., Bosmans, J. H., Drost, N., ... & Bierkens, M. F. (2018). PCR-GLOBWB 2: a 5 arcmin global hydrological and water resources model. *Geoscientific Model Development*, *11*(6), 2429-2453.

Line 411: I do not see a general improvement of discharge. It looks a bit random to me. How do you come to the conclusion that the performance is 'decent overall'. Please quantify.

**Response:** Thank you for pointing this out. We removed the phrase "although performance is decent overall" from the text as this is subjective and quantified the discharge performance in the previous sentence instead.

Line 426: I am not sure I can see this improvement in the Fig. 5 G and H. Looking at Fig. G and H I do not see any improvement in model performance with increasing complexity of the snow routine.

**Response:** Thank you for making us aware of this. Indeed, there is no improvement for the observational stations in 5G and H. We have now explicitly mentioned this in the text, and at the same time added an explicit quantification of the regional performance changes per elevation zone.

*"Without calibration, further structural changes to the snow module (i.e. the seasonally varying DDF, exponential temperature dependence, and a rain-to-snowfall transition temperature range) **did not improve performance against observational SWE stations (see Figure 5G and H). Averaged over elevation zones, however, these changes** lead to an improvement of the SWE representation at higher elevation…" (L469-472)*

Line 430: I do not see the evaluation of 'melt rates'. You calibrate a DDF which is the same for all the area. How are there different melt rates depending on elevation? Please explain.

**Response:** Indeed, we don't directly evaluate melt rates. What we mean by melt rates is the amount of snow melted per day (with unit m/day), which is different from the DDF (with unit m/day/deg C). As the DDF varies with the season and snowmelt timing varies with elevation, this leads to elevation-dependent differences in the melt rates at the same temperatures. Therefore, we conclude that the differences in modelled SWE capture such elevation-dependent patterns.

We have further clarified this in the text:

*"These differences between elevation zones suggest that the model structure **leads to varying melt rates with elevation: this is to be expected, as the temperatures at which snow starts to melt are reached later in the year at higher elevations, so that the same temperatures are combined with different values of the time-varying DDF which produces differing melt rates."** (L476-478)*

Line 462: Please explain the performance decrease in the south-western Switzerland.

**Response:** We clarified in the text that such performance changes are likely due to reservoir regulation:

*"Furthermore, our results indicate that in certain regions, especially around the **heavily-regulated** Rhône river in south-western Switzerland, discharge performance can decrease with the addition of glaciers (see Figure 6E). Such performance decrease generally occurs in catchments with a negative **WB** or reservoir **regulation** ...." (L510-512)*

Line 491: 'These changes have similar magnitudes as the changes due to different forcing data'. This is an interesting sentence, as you previously state that the forcing data has a very strong influence. Does this mean also the selection of the evaluation period has a strong influence?

**Response:**  Whereas changes for individual catchments indeed depend on the chosen evaluation period, performance over the whole domain does not vary much with the choice of evaluation period (see Figure 11F).
We have clarified this in the text:

*"These changes have similar magnitudes as the changes due to different forcing datasets (compare to Figure 4G,H, and I**), suggesting that the selected evaluation period is important for model performance in individual catchments. For the domain as a whole,** the warmer Evaluation II period shows a slight drop in median performance..." (SI L: 9-14)*

Note that this text on the Transferability analysis has been moved to the Supporting Information as mentioned before.

Line 494 and Line 630: 'Transferability to warmer climate conditions': I do not think that the analysis at the current state sufficiently supports this statement. You use an highly empirical approach for snowmelt and calibrate it to a specific time period. I do not see how the comparison of 10-year time slices can prove that the DDFs will be the same end of the century. In my opinion, the discussion of the transferability of DDFs in time also is not the focus of the study. As there are a lot of other interesting aspect to focus on, please consider to shorten/revise/remove this part.

**Response:** We agree that this is not the main focus of this study. Therefore, we moved the "Transferability" section to the Supporting Information and removed or weakened any claims related to this section in the text:

In Discussion:
*"**A short** evaluation, which explicitly addressed model transferability, shows that model performance for discharge and SWE remains mostly consistent over the warmer and colder evaluation periods compared to the reference period, **although the temperature changes over the study period are limited (Figure S1 in the Supporting Information).**" (L667-669)*

---------------------- **Review 2** --------------------------------------------------------------------------------

RC2: **'Comment on egusphere-2024-3072'**, Kristian Förster, 04 Mar 2025

In their manuscript "Hyper-resolution large-scale hydrological modelling benefits from improved process representation in mountain regions" Joren Janzing et al. describe a set of model extensions to PCR-GLOBWB 2.0 in order to make it more suitable for kilometre-scale

modelling in mountainous regions. Together with model improvements in representing snow, glaciers, and soil, different forcing datasets are also tested. In essence, they found that the new model is capable of representing relevant processes in mountain hydrology quite well, even though the original model has been developed for large scale analyses with coarser spatial resolution. The manuscript fits well into the scope of Hydrology and Earth System Sciences. I found that the analyses carried out in the manuscript clearly show the added value of the most recent advances even though I think that some points require some more explanations and / or discussions. Apart from that I believe that the manuscript is an important contribution and I am looking forward to reading the final revised manuscript. Please find my comments below.

**Response:** Thank you for assessing our work and for your careful and constructive feedback. We address your suggestions and concerns below.

As a general point, we wanted to point out that we re-executed some of the model runs, which led to some slightly different results than in the first draft. We also noticed there were a few gauges which fell outside of our chosen region (e.g. the upstream part of the catchments of the Danube, Rhone, Rhine and Po), which is why we reduced the number of references gauges.

General comments

1.   Structure of experiments / evaluations. The order of single steps is different throughout the manuscript. Abstract: forcing, snow, glaciers, soil; Introduction: snow and glaciers, soil, forcing. Hypotheses: snow and glaciers, "reviewing standard parameterizations" (I think that H2 should be rephrased in order to better reflect what was actually done. Do you mean reviewing and adjusting?), forcing. Methods and later results: Forcing, snow and glaciers, and soil. Conclusions: Snow and glaciers, soil, forcing. I was wondering if it would help the readers to agree on a consistent order of steps throughout the manuscript?

**Response:** Thank you for highlighting the need for more consistency in the order of the experiments. We have followed your suggestions and improved consistency by generally following the order "Forcing;  Snow; Glaciers; Soil" - this aligns with the order of our experiments. We have changed this in the conclusions and the hypotheses, meaning that the numbers of the hypotheses have changed.

We make one exception and deviate from this rule in the introduction. The reason for this is that we consider the snow and glacier component to be more important for our storyline than the meteorological forcing.
We have implemented the following changes in the text to clarify why the specific order was chosen (note that we have also changed the hypothesis regarding the standard parameterization in accordance with your suggestion).

In section Introduction:
*"**Nevertheless, any potential gains in model performance due to improved process representation or parameterization are constrained by the quality of the model input. It is thus important that such conditions are met first. H**ydrological modelling is indeed sensitive to the meteorological forcing dataset used as input…"* (L116-118)

*"While large-scale hydrological simulations at higher spatial resolution have become feasible thanks to increasingly available computational resources, it is yet unclear by how much hydrological simulations can improve when combining such high-resolution models **with the latest generation of meteorological forcing datasets and improved process representation.** Therefore, we here aim to explore the effect of **(1)** using different meteorological datasets, **(2)** improving snow and glacier representations, and **(3)** changing runoff partitioning in the soil **on discharge simulations in PCR-GLOBWB 2.0.**" (L133-137)*

*"We hypothesize that hyper-resolution LHM performance for discharge in mountain regions will increase by **(H1)** using forcing products that include a representation of smaller-scale atmospheric dynamics compared to other forcing products, **(H2)** improving the representation of mountain hydrological processes, such as snow and ice melt; and **(H3)** reviewing **and adjusting** standard parameterizations" (L141-144)*

In section Conclusion:
*"– Meteorological forcing is uncertain over Alpine regions and different forcing datasets lead to major differences in hydrological model performance for discharge. These performance differences vary spatially between different forcing datasets.*
*– Discharge simulations forced by a reanalysis product using high-resolution atmospheric dynamics at 5.5 km (CERRA-CHELSA) did not consistently outperform simulations with the other forcing products which use coarser atmospheric dynamics at 31 km (STANDARD and CHELSA; **Hypothesis 1** is not supported).*
*– An improved representation of snow and glaciers improves SWE and discharge simulations in high Alpine catchments where natural flow conditions dominate (**Hypothesis 2** is supported).*
*– The introduction of better runoff partitioning in the soil leads to an improvement of discharge simulations in smaller rain-dominated catchments by increasing their flashy response to rainfall peaks. However, these catchments still show overall weak model performance for discharge, suggesting that there is more to gain from improved routing in the soil (**Hypothesis 3** is supported)." (L692-703)*

1. Water balance signature in L371 (and also Figure 7): I found the adoption of the Water balance signature not well introduced into the evaluation even though it seems to be a reasonable indicator. Firstly, I would recommend to better explain what positive and negative values mean. From what I could get from the equation (and later by referring to Salwey et al., 2023), it seems to me that a positive value suggests a gain in water, while negative values would indicate a loss of water. Is that true? Please consider adding a short description that could help the readers to better follow your interpretation later. Secondly, I was wondering why you call this water balance signature later in the results section "water gap"? This even more confusing, since positive values seem to be related to additional water rather than a gap. I would suggest to refer to the original term suggested by Salwey et al., 2023.

**Response:** Thank you for these suggestions. We have changed our terminology to "Water balance signature" to match the terminology used in Salwey et al.,2023 and later in the text mostly refer to it as WB. Furthermore, we have added additional clarification of how this metric should be interpreted.

Textual changes:

*"**Another way** to separate regulated from natural catchments **is the water balance signature (WB), which describes** the normalized deviation from a closed water balance assuming no long-term storage effects (Salwey et al., 2023)."* (L400-402)

*"**Positive values of WB indicate that a catchment's discharge is higher than expected. Assuming that the meteorological components could be well-estimated, such positive values could suggest that a catchment gains more water than what comes in through precipitation. Negative values of WB indicate that a catchment loses more water than just the potential evapotranspiration.**"* (L405-408)

*"**However,** other factors such as errors in the meteorological forcing or additional water input from glaciers **due to imbalance** can also lead to strong water balance deviations, which can also affect model performance with respect to discharge. **We thus use WB as a general metric to study the effect of such deviations in the water balance on model performance. In addition, we also use WB as an indication for water transfers, hydropower production and other water balance deviations in combination with catchment-based information on reservoirs.**"* (L411-415)

1.  Glaciers in Figure 8: I agree that your modelling approach might be more suitable to regions of glaciers rather than individual ones. However, I really went through hard times reading Figure 8. Panels a) and b) are hard to read. Would be worth to increase the zoom level and to make glacier outlines smaller? Where is Mer de Glace (L447) on the map? I would suggest updating Figure 8 a) to d) in order to improve readability. The larger glaciers seem to have a positive mass balance bias, see panels e) and f), while in panel g) the model seem to have quite good average skill for glaciers with an area of more than 8 square kilometers. Why do we see a dramatic depletion in skill when looking at smaller glaciers? Why is 8 square kilometers chosen here? Is it related to the threshold in the Delta h method, proposed by Huss et al. (2010)? I was wondering if a regional mass balance, similar to e) and f), would better support your findings? Moreover, what remains unclear is whether the mismatch in glacier response time is related to differences in model (structure) or related to different forcings, when compared to Zekollari et al. (2020)?

**Response:** Thank you for making us aware that Figure 8 is indeed not easy to understand. We have improved Figure 8 by implementing your helpful suggestions, such as further zooming in on the considered glaciers, showing thin outlines and highlighting some specific glaciers.

Regarding the additional points you listed:
- Mass balance bias of larger glaciers: we highlighted the Aletsch glacier and Glacier d'Argentiere in plots G, to indicate that there is indeed some bias. Note however that the mean mass balances are only shown over the Evaluation period II and that we have changed this subplot to show geodetic mass balances.

- Depletion of skill for smaller glaciers and for response times compared to Zekollari et al., 2020: Indeed, there is a significant drop in performance for smaller glaciers. We touched upon the possible reasons for this in the Discussion section.

*"Still, we note that glaciers can show significant biases in the mass balance (see Figure 8G) and in the responses (see Figure 8H). Differences in long-term responses compared to previous experiments in which ice dynamics were incorporated might be partly*

*explained by the Δh-parameterization, which ignores potential delays in glacier response to mass changes (Seibert et al.,2018a). Further biases in both mass balance estimates and responses are likely related to the relatively coarse spatial resolution of our model compared to the size of individual glaciers, which makes it more difficult to accurately describe patterns in melt or snow accumulation for small glaciers consisting of only a few grid cells. Melt representation of individual glaciers could be improved by resolving glaciers at higher spatial resolution than the resolution simulated here, for example by including elevation zones (Seibert et al., 2018a). Resolving glaciers at higher spatial resolution could also lead to even more realistic glacier retreat, since the Δh-parameterization was originally designed for higher spatial resolutions (Huss et al., 2010) and is likely less accurate at the coarser 30-arcsec model grid." (L590-600)*

- Choice for 8 km$^2$: We had chosen this threshold somewhat arbitrarily to illustrate the effect of glacier size on performance, as we noted a clear relationship between the two. We have now removed this sharp glacier area division and instead show a more continuous relationship between model performance and glacier area to make it less arbitrary.

- Regional mass balance: Thank you for this interesting suggestion. We have considered adding a subplot of a time series of the regional mass balance, similar to 8E and F, but in the end decided against this as this approach would include many additional assumptions. The resulting time series would heavily depend on our selection of glaciers - of which only a few have (long term) yearly measurements and which might not be equally distributed in space and time.
Instead, we recreated subplot 8G based on geodetic measurements from Hugonnet et al., 2021, which have full spatial coverage and thus allow us to have regional estimates of the mass balance but only over a 10-year time period. Furthermore, we have changed the plot of the response times to the long term response simulations to highlight the more general behaviour of larger groups of glaciers together.

Hugonnet, R., McNabb, R., Berthier, E., Menounos, B., Nuth, C., Girod, L., ... & Kääb, A. (2021). Accelerated global glacier mass loss in the early twenty-first century. *Nature*, *592*(7856), 726-731.

1.  Glacier initialization in 1990. How do initialize glaciers (area, volume) in your model setup when you deviate from the calibration period (which coincides with the data in Farinotti et al., 2019 for the early 2000)?

**Response:** We indeed used the consensus estimates from Farinotti et al., 2019, which are representative for the early 2000s, to initialize our glaciers in 1990.  We agree that this may lead to inconsistencies in glacier outlines around 2000. Therefore, we changed the implementation and keep  glacier volumes and outlines constant until 2000, although we still calculate ice melt and accumulation. This new approach is largely similar to previous large-scale hydrological modelling studies (e.g. Hanus et al., 2024). We acknowledge that  this updated procedure is still not optimal. However, to our knowledge, there are no glacier volumes/ice thicknesses available representing conditions before 1990 at the global and continental scale (needed to easily apply the model to other regions).

We have added some further clarification of the new approach in the text:

In section Datasets:

**"To implement a new glacier module, we used the locations of the existing glaciers and their initial ice thickness using the consensus estimate from Farinotti et al. (2019), which are representative for the year 2003 for most glaciers. To our knowledge, there are no estimates of glacier volumes available that go further back in time and that cover all glaciers on the global, continental or Alpine scale. This glacier volume dataset has a varying spatial resolution (max. 200 m) has previously also been used to initialize glaciers for the same study period by Hanus et al. (2024)."** (L190-194)

In section Glacier module

*"To account for such temporal changes, we implement the empirical Δh-parameterization scheme from Huss et al. (2010), as implemented in HBV (Seibert et al., 2018a).* **As the glacier volumes are representative for around the year 2003, we only apply this scheme after the year 2000, keeping the glacier volume and area constant between 1990-2000 while still calculating ice melt and accumulation."** *(L303-307)*

Farinotti, D., Huss, M., Fürst, J.J., Landmann, J., Machguth, H., Maussion, F., Pandit, A., 2019. A consensus estimate for the ice thickness distribution of all glaciers on Earth. Nature Geoscience. https://doi.org/10.1038/s41561-019-0300-3

Hanus, S., Schuster, L., Burek, P., Maussion, F., Wada, Y., & Viviroli, D. (2024). Coupling a large-scale glacier and hydrological model (OGGM v1. 5.3 and CWatM V1. 08)– Towards an improved representation of mountain water resources in global assessments. *Geoscientific Model Development, 17*(13), 5123-5144.

Specific comments:

L15: What do you mean by unidirectional?
**Response:** We clarified that we mean that changes can be both positive and negative:

*"Our evaluation of the effect of these different adjustments on model performance for discharge shows that while the meteorological forcing has a major effect on discharge simulations,* **it results in a mixed pattern of performance gains and losses over the domain."** *(L13-15)*

L28: "thanks to heatwave". In my opinion you should highlight that this was only possible through a very negative mass balance.
**Response:** We have addressed this point by highlighting in the text that the mass balances were very negative.
*"...whereas Alpine glaciers provided surrounding rivers with surplus melt-water during the 2003 Central European Drought* **due to very negative mass balances** *(Van Tiel et al., 2023)." (L28-29)*

L60pp.: I think that there are lots of other references that support your statement here.
**Response:** Thank you for pointing out the need to include additional literature to support this statement. We have added several references to support this statement, and have also specified that with "a constant DDF" we refer to a "time-constant DDF".

*"However, LHMs often use very simplistic temperature-index schemes e.g. by using only **a time-constant** DDF or by omitting calibration **(e.g. Gosling and Arnell, 2011; Sutanudjaja et al., 2018; Müller-Schmied et al., 2021; Stacke and Hagemann, 2021)**." (L60-62)*

Gosling, S. N., & Arnell, N. W. (2011). Simulating current global river runoff with a global hydrological model: model revisions, validation, and sensitivity analysis. Hydrological Processes, 25(7), 1129-1145.

Müller Schmied, H., Cáceres, D., Eisner, S., Flörke, M., Herbert, C., Niemann, C., ... & Döll, P. (2021). The global water resources and use model WaterGAP v2. 2d: Model description and evaluation. Geoscientific Model Development, 14(2), 1037-1079.

Stacke, T., & Hagemann, S. (2021). HydroPy (v1. 0): A new global hydrology model written in Python. Geoscientific Model Development, 14(12), 7795-7816.

Sutanudjaja, E. H., Van Beek, R., Wanders, N., Wada, Y., Bosmans, J. H., Drost, N., ... & Bierkens, M. F. (2018). PCR-GLOBWB 2: a 5 arcmin global hydrological and water resources model. Geoscientific Model Development, 11(6), 2429-2453.

L74pp.: Please explain where exactly theses percentage values refer to.
**Response:** We have further clarified to which gauging stations these percentages refer to:

*"**.. on average, glacier storage change contribution to total runoff ranges from 4% in the river Danube at Ceatal Izmail in September to 25% for the river Rhône at Beaucaire in August (Huss, 2011), …"** (L76-78)*

L77pp.: Introducing one-way coupling would also suggest that two-way-coupling exists, too (see Pesci et al., 2023).
**Response:** We have removed the specification "one-way" to acknowledge this, as the general set-up (using glacier output into the model) is still the same.

*"b.) using the output from an external glacier model as the input to the hydrological model (i.e.  coupling the models; "coupled models")." (L82-83)*

L113. I felt some statement is missing here before introducing meteorological forcing.
**Response:** Thank you for your suggestion. We have added a sentence to smoothen the transition from the previous paragraph to the one on meteorological forcing:

*"**Nevertheless, any potential gains in model performance due to improved process representation or parameterization are constrained by the quality of the model input. It is thus important that such conditions are met first.** Hydrological modelling is **indeed** sensitive to…."* (L116-118)

L135: Please consider rephrasing H2 (as suggested before).
**Response:** Following your earlier comment, we have changed H2 in the way you suggested (it is H3 now):
*"…**(H3)** reviewing **and adjusting** standard parameterizations." (L144)*

L176: I am not sure whether "to guarantee" is correct in this context.
**Response:** Thank you for highlighting this. We have changed the word "guarantee" to "generate more":

*"In contrast, near-surface air temperature can be further downscaled to **generate more** accurate spatial melt patterns at 30 arcsec resolution." (L185-186)*

L239: DDF in general depend on land use. For the large-scale model it's clear that working with an "average" DDF makes sense. However, I remember from own research that DDF shows a quite different behavior in forests. Given the increased resolution and the relevance of forests in the water balance in mountainous areas, I think it is at least worth to discuss this later.

**Response:** We have expanded our discussion on spatial patterns in DDFs by explicitly discussing the potential role of land cover and vegetation.

In section Discussion:
*"Another reason for this slight decrease in discharge performance at lower elevations could be related to our choice of regionally-averaged DDFs, since in reality these DDFs show smaller-scale variability in space (e.g. with aspect, albedo, elevation, **landcover, vegetation,**...)* ***(Kuusisto, 1980; Rango and Martinec, 1995;*** *Hock, 2003; Ismail et al., 2023****)" (L566-569)***

*"More elaborate snow module formulations, such as parameterizations that include aspect (e.g. Immerzeel et al., 2012) or radiation (e.g. Hock, 1999), could increase our ability to capture more detailed spatial melt patterns.* ***Such approaches can become feasible now that higher spatial resolutions can resolve slopes and represent vegetation cover and land use in a more detailed way.*** *However, they come at the cost of increased model complexity and a larger number of input variables." (L570-574)*

Kuusisto, E. (1980). On the values and variability of degree-day melting factor in Finland. Hydrology Research, 11(5), 235-242.
Rango, A., & Martinec, J. (1995). Revisiting the degree-day method for snowmelt computations 1. JAWRA Journal of the American Water Resources Association, 31(4), 657-669.

L261p: I do not understand that snow is transported to glaciers, given that transport is only considered to non-glaciated cells. Could you please explain this please?
**Response:** Thank you for highlighting the need for further clarification. Based on this comment and comments from the other reviewers, we have added further details:

*"However, the snow* ***that is transported*** *should sometimes be part of glacier accumulation, which is why we here apply the lateral transport scheme only outside of glaciers. When we introduce the glacier module, we thus restrict the lateral snow transport and apply it only on non-glacierized areas. This means that snow can* ***only*** *be transported* ***from a.) a non-glacierized cell to a non-glacierized cell and b.) from a non-glacierized cell to a glacierized cell. There is no snow transport from a glacierized cell to either a glacierized cell or a non-glacierized cell. When snow is transported onto a glacierized cell, it is becomes part of the snow cover on the glacier: it can thus reduce ice melt and become part of the glacier accumulation (Kuhn, 2003; Freudiger et al., 2017)." (L279-285)***

Freudiger, D., Kohn, I., Seibert, J., Stahl, K., & Weiler, M. (2017). Snow redistribution for the hydrological modeling of alpine catchments. *Wiley Interdisciplinary Reviews: Water*, *4*(5), e1232.
Kuhn, M. (2003). Redistribution of snow and glacier mass balance from a hydrometeorological model. Journal of Hydrology, 282(1-4), 95-103.

L271: Do you mean lower albedo?
**Response:** Thank you for making us aware of this inconsistency. Indeed, we meant lower albedo and have changed this in the text.

L294: If glaciers can only grow to its original extent, how do you realize to initialize the glaciers with a larger extent in the past?
**Response:** As pointed out above, this is a severe limitation caused by the lack of glacier volumes available further back in time. We thus keep the glacier volume constant until 2000, when we assume the volumes are the same as in the observed datasets.

We refer to the same added text as mentioned above:

*"To implement a new glacier module, we used the locations of the existing glaciers and their initial ice thickness using the consensus estimate from Farinotti et al. (2019), which are representative for the year 2003 for most glaciers. To our knowledge, there are no estimates of glacier volumes available that go further back in time and that cover all glaciers on the global, continental or Alpine scale. This glacier volume dataset has a varying spatial resolution (max. 200 m) has previously also been used to initialize glaciers for the same study period by Hanus et al. (2024)."* (L190-194)

L311: I am sorry if I missed this important detail but for me it's not clear if you calibrate a single parameter (e.g., DDFmin) for the entire domain or catchment-wise. From the table in supplement, it seems to me that it's a single value only which is valid for the entire domain after calibration.
**Response:** Thank you for raising this point. Indeed, we calibrate these parameters for the domain as a whole. We added some additional clarification.

*"We calibrate on the specific process (e.g. on SWE) instead of discharge to avoid compensating for deficiencies in other processes with parameter calibration and to increase the stability of these parameters when temperature change (Sleziak et al., 2020). We try to find one parameter set that is regionally valid, **i.e. constant over the entire domain.**"* (L337-339)

L378: Why is snow calibrated only in Switzerland (I hope that I understood it correctly)? Here you focus on both Switzerland and Austria?
**Response:** Thank you for raising this point. We clarified that we do this because we are interested in testing the transferability of the model to other regions, as we do not have comparable snow products for other parts of the Alps such as Italy or France.

In section Calibration:
*"However, as every other SWE dataset, this product also comes with some uncertainties. **Note that we do not calibrate over Austria, even though we have similar data available there as well, to be able to test the transferability of the snow scheme to other regions.** To explicitly account for elevation-dependent melt patterns,…"* (L349-352)

In section Evaluation:

*"In addition to discharge, we evaluate model performance for SWE by calculating spatial averages over different elevation bands (0-1000 m, 1000-2000 m, 2000-3000 m or the entire country) over Switzerland. **Additionally, we evaluate it over Austria, which has not been used for model calibration and can therefore provide insights on how well the model generalizes to other regions.**"* (L416-419)

L383: How do you evaluate glacier geometry evolution? Later you explain that this was only done visually.
**Response:** Thank you for raising this point. We have highlighted that we evaluate the glacier geometry evolution both visually to check the plausibility of spatial patterns and over longer time intervals compared to Zekollari et al., 2020**.**

*""To evaluate glacier **geometry evolution,** we both visually compare spatial patterns of glacier surface elevation changes against observations (GECM; see Table 2) **and quantitatively evaluate long-term glacier changes**. It is difficult to evaluate the long-term response of simulated glaciers to climate forcing given the relatively short study period."* (L426-428)

Zekollari, H., Huss, M., Farinotti, D., 2020. On the Imbalance and Response Time of Glaciers in the European Alps. Geophysical Research Letters 47, e2019GL085578. https://doi.org/10.1029/2019GL085578

L442p: Here, I think it would be interesting to provide a few more details what we see, e.g., the values around zero for small snow fractions. By the way: Is snow fraction the percentage of precipitation that is solid or how is it defined?

**Response:** Based on your comment and comments from other reviewers, we have provided some further details about the results and also indicated in the text that snow fraction indeed refers to the percentage of precipitation that is solid. We have further clarified this by using the term "snowfall fraction" and adding an explanation to the text.

*"Figure 7A, B, and C illustrate that the **changes in the snow module** mainly **improve** model performance for discharge **in catchments with high snowfall fractions, whereas in catchments with low snowfall fractions the changes are negligible or slightly negative. Still, some catchments with high snowfall fractions experience decreases in performance for discharge: these decreases mostly happen in the presence of reservoirs and/or negative values of WB …"** (L491-494)*

L491pp: When comparing the drop in median for the different periods, it would be relevant to mention them in the text. Given the very good representation of SWE in the warmer period, is there any observation regarding the glaciers?

**Response:** Thank you for your comment. Based on comments from Reviewer 1, we have decided to place less focus on this section and moved the Figure to the Appendix. We added a subplot on comparing glacier performance between the different periods. We also added a quantification of the changes to the text in the Supporting Information.

Figure 12: Adding KGE or some other score would be helpful. Why does the full model compute a very high peak for Massa. Given that it is a 10 yrs. average, it seems to be a quite high event.

**Response:** We have added a KGE score for the benchmark and the final run to the figure.

We have further investigated the peak for the Massa river and it appears to be related to the modelling choice that we converted all the snow to glacier ice on the same day (i.e. beginning of the hydrological year), which led to increased melt around that day. We implemented new runs with an updated glacier module which removes the snow continuously, similar to the original method by Seibert et al., 2018, which does not have this problem.

Seibert, J., Vis, M. J., Kohn, I., Weiler, M., & Stahl, K. (2018). Representing glacier geometry changes in a semi-distributed hydrological model. *Hydrology and Earth System Sciences*, *22*(4), 2211-2224.

Technical comments:

Figure 4: Unfortunately, the legend hides important details of the distribution. Would it be possible to make its background transparent? See also Figure 5, 6, 11

**Response:** Thank you for the suggestion. We have moved the legend outside of the distributions plot for all suggested figures.

Figure 10: KGE refers to the upper row A-C, while KGESS to the bottom row (D-F)?

**Response:** That is indeed correct. We have further highlighted this in the Figure caption.
*"Model performance of the Full model run for discharge (top row**, showing KGE**) and the total model performance changes for discharge compared to the CERRA-CHELSA benchmark run*

*(bottom row**, showing KGESS**) in the evaluation catchments in relation to different catchment characteristics."* (caption Figure 10)

References

Farinotti, D., Huss, M., Fürst, J.J., Landmann, J., Machguth, H., Maussion, F., Pandit, A., 2019. A consensus estimate for the ice thickness distribution of all glaciers on Earth. Nature Geoscience. https://doi.org/10.1038/s41561-019-0300-3

Huss, M., Jouvet, G., Farinotti, D., Bauder, A., 2010. Future high-mountain hydrology: a new parameterization of glacier retreat. Hydrology and Earth System Sciences 14, 815–829. https://doi.org/10.5194/hess-14-815-2010

Pesci, M.H., Overberg, P.S., Bosshard, T., Förster, K., 2023. From global glacier modeling to catchment hydrology: bridging the gap with the WaSiM-OGGM coupling scheme. Frontiers in Water 5, 1296344. https://doi.org/10.3389/frwa.2023.1296344

Salwey, S., Coxon, G., Pianosi, F., Singer, M.B., Hutton, C., 2023. National-Scale Detection of Reservoir Impacts Through Hydrological Signatures. Water Resources Research 59, e2022WR033893. https://doi.org/10.1029/2022WR033893

Zekollari, H., Huss, M., Farinotti, D., 2020. On the Imbalance and Response Time of Glaciers in the European Alps. Geophysical Research Letters 47, e2019GL085578. https://doi.org/10.1029/2019GL085578

---------------------- **Review 3** --------------------------------------------------------------------------------

**RC3:** **'Comment on egusphere-2024-3072'**, **Anonymous Referee #3, 05 Mar 2025**

This study describes the implementation of several modifications to an existing Large Hydrological model and evaluates the effects of implementing those changes stepwise. Specifically, it evaluates the implementation of a snow transport scheme, an altered snow model parametrization, improvements related to calibration of that snow model, a new glacier model, and altered soil parameters. The authors also evaluate model differences related to meteorological forcing uncertainty.

**General comments:**

Overall this is a well-organized paper that clearly tracks the series of implemented changes. The authors justify their choices of evaluation data and techniques, and provide sufficient evidence for most conclusions. However given the complexity of the study, there are a few places where further clarification, justification, or tempered conclusions are needed.

**Response:** Thank you very much for the time and effort you put into reviewing our manuscript and your detailed comments and feedback. We have responded to your questions, suggestions and concerns below.

As a general point, we wanted to point out that we re-executed some of the model runs, which led to some slightly different results than in the first draft. We also noticed that there were a few gauges which fell outside of our chosen region (e.g. the upstream part of the catchments of the Danube, Rhone, Rhine and Po), which is why we reduced the number of references gauges.

The only substantial piece of additional justification pertains to Figure 7 and your conclusions of performance in non-regulated versus regulated catchments. While the correlation of performance with water gap sign is quite clear for the second and third row, I'm not totally convinced how well the value of the water gap fraction works in identifying natural vs regulated catchments (to my eye both improvements and deterioration of performance are pretty evenly split between those with and without reservoirs marked as + and o). This is important because you later equate locations with performance improvements to natural catchments and locations with deterioration to regulated catchments (Table 3 and around line 565 and 606). Can you provide any additional justification for this association over your domain? Since the Salwey study was over Great Britian it may not transfer well to more inland mountainous locations. How do you know that another variable isn't controlling the relationship between WB and performance improvement? For example, maybe improvements in the model occur at locations that correspond to thin soil in the real world and where the hydrological response is flashier (since you effectively biased your model to better performance at such locations). Such locations might have limited capacity to store water longer term which would correlate with WB and could appear as a signal in non-regulated catchments.

**Response:** Thank you for making us aware that additional justification is needed. Based on this comment and comments by the other reviewers, we have implemented several changes.

1. The WB signature is indeed a complicated metric and is influenced by many other processes. We have now rewritten the text to better indicate that we do not assume that the WB signature corresponds one-on-one to reservoir regulation and to better highlight that reservoirs are just one process that can affect this signature. We also explicitly derived conclusions for reservoirs only by combining information on the WB metric and data on reservoirs. We have made several changes to the text:

*"**Another way** to separate regulated from natural catchments **is the water balance signature (WB), which describes** the normalized deviation from a closed water balance assuming no long-term storage effects (Salwey et al., 2023)."* (L400-402)

*"**Positive values of WB indicate that a catchment's discharge is higher than expected. Assuming that the meteorological components could be well-estimated, such positive values could suggest that a catchment gains more water than what comes in through precipitation. Negative values of WB indicate that a catchment loses more water than just the potential evapotranspiration."** (L405-408)*

*"**However,** other factors such as errors in the meteorological forcing or additional water input from glaciers **due to imbalance** can also lead to strong water balance deviations, which can also affect model performance with respect to discharge. **We thus use WB as a general metric to study the effect of such deviations in the water balance on model performance. In addition, we also use WB as an indication for water transfers, hydropower production and other water balance deviations in combination with catchment-based information on reservoirs."** (L411-415)*

2. We strengthened the evidence for a connection between WB and the reservoirs. We added a new Figure i to the Supporting Information that shows that WB is related to a high degree of reservoir influence (which we here define as the total reservoir volume per catchment divided by the annual average discharge). We also redefined when we consider a catchment to be regulated, namely that we define this based on the degree of regulation. We only consider catchments where the degree of regulation exceeds 0.1 days to be regulated, which removes catchments with reservoirs but hardly any regulation.

[Figure]

*Figure i Relationship between the degree of reservoir regulation and the WB signature. The degree of reservoir regulation is here defined as the total reservoir volume over the catchment divided by the mean annual discharge.*

**Other Specific comments (generally minor):**

Line 37: "regional to local-scale" can be interpreted differently. Please be more explicit.
**Response:** Thank you for pointing this out. We adjusted the text:

*"... limits the usefulness of their output for policymakers, who are often interested in **more detailed information.**" (L38-39)*

Figure 1: The range of colors on the map doesn't look like it fully matches those on the color bar (which doesn't seem to have the darker greens and bluer greens). Is there some sort of transparent overlay of other colors on the map?
**Response:** Thank you for pointing this out. That is correct, since we wanted to highlight the most important river basins on the map. We have updated the figure to show the topography without the shading and included subplots that show the subbasins, as well as the countries and discharge stations.

Line 160: Since the precipitation isn't downscaled to the resolution of your model, it might be clearer to specify "(3) CERRA-CHELSA, a mixed dataset with temperature from the Copernicus European Regional ReAnalysis (CERRA) further downscaled using the CHELSA algorithm and precipitation data directly from CERRA-Land."
**Response:** Thank you for your suggestion. We have implemented the suggested specification in the text.

Line 180-221: This section needs some clean-up to help clarify details. You mix in both data required for evaluation and ancillary data required to run the model (e.g. RGI) or produce metrics (e.g. snowfall fraction, PET). It's hard to sort out what is what, especially at a first read. In some cases, datasets mentioned don't appear in the Table (e.g. RGI, GLAMOS) and in others they appear in the table but aren't discussed here (Farinotti glacier volumes). You might try splitting up the discussion of strictly evaluation data versus ancillary data needed to run the model or compute metrics.
**Response:** Thank you for making us aware that this section needed an improved structure and a consistency check. We followed your suggestion and made a clearer split between input, evaluation and ancillary data.

We restructured the text and made this split more apparent. The section is now called "Datasets", with the subsections "Model input", "Evaluation and calibration data" and "Ancillary data". We reordered the information accordingly.

Furthermore, we added the RGI and GLAMOS to the Table and added a description of the Farinotti dataset to the text in this section as well:

**"To implement a new glacier module, we used the locations of the existing glaciers and their initial ice thickness using the consensus estimate from Farinotti et al. (2019), which are representative for the year 2003 for most glaciers. To our knowledge, there are no estimates of glacier volumes available that go further back in time and that cover all glaciers on the global, continental or Alpine scale. This glacier volume dataset has a varying spatial resolution (max. 200 m) has previously also been used to initialize glaciers for the same study period by Hanus et al. (2024)."** (L190-194)

Lines 237-251: What values were chosen for $m\_m$ and $m\_p$? Do these not alter the calibration? - at line 320 you state that you only calibrate the two DFF values.
**Response:** Thank you for pointing out the need for further clarification. We further specified that these two parameters were taken from the study by Magnusson et al., 2014. They derived the $m\_p$ value from measurements and they fixed the $m\_m$ value to reduce the number of parameters for calibration.
These parameters can be found together with the other parameters used in the Supporting Information in Table S1.

In section Model development:
*"The parameters used in the equations **and any fixed values** are listed in Table S1 **in the Supporting Information**."* (L243-244)

In section Snow module:
*"… where $m\_m$ is a parameter controlling the transition between melt and no melt (∘C) **and was kept constant by Magnusson et al., (2014)."*** (L264-265)
*"…and the parameter $m\_p$ determines the range where snow and rainfall co-occur (∘C) **and was derived from snowfall observations by Magnusson et al., (2014)**."* (L270-271)

In section Calibration:
*"In contrast, model evaluation is performed using the full model run. The considered calibration period is 2000–2009 (see Section 2.4). **Note that any other parameters not mentioned here remain fixed and their values can be found in Table S1 in the Supporting Information."***
*(L344-345)*

Magnusson, J., Gustafsson, D., Hüsler, F., & Jonas, T. (2014). Assimilation of point SWE data into a distributed snow cover model comparing two contrasting methods. Water resources research, 50(10), 7816-7835.

Lines 255-270: Please reword to clarify the interactions between snow and glaciers regarding both lateral transport, accumulation from snowfall, and melt. Including an arrow for lateral transport in Figure 3 may help. In particular I think it would help to more explicitly describe how the model treats the three possible cases: transport "onto" glaciers (i.e. non-glacier to glacier; I think this is what you have implemented and focus on), transport "off" glaciers (i.e. glacier to non-glacier; maybe this is what you restrict from occurring?), but also clarify whether lateral transport still occurs from non-glacier to non-glacier cell. On glaciers, is the laterally transferred snow considered as a separate source from the snow accumulation from snowfall? For example, is laterally transferred snow converted to glacier ice based on its mass but the snow accumulated from snowfall sits "on top" of the glacier and must melt off? ("the glacier

only melts when it is not covered by snow"). Or are the two sources of snow put into the same reservoir which can only convert to glacier ice on sept 1? (in which case I guess there's no glacier melt that season)

**Response:** Thank you for highlighting the need to better clarify the different cases of transport. We have indicated in Figure 3 how lateral snow transport is added to the snow reservoir on the glacier and have implemented the given suggestions in the text in the following way:

*"However, the snow **that is transported** should sometimes be part of glacier accumulation, which is why we here apply the lateral transport scheme only outside of glaciers. When we introduce the glacier module, we thus restrict the lateral snow transport and apply it only on non-glacierized areas. This means that snow can **only** be transported **from a.) a non-glacierized cell to a non-glacierized cell and b.) from a non-glacierized cell to a glacierized cell. There is no snow transport from a glacierized cell to either a glacierized cell or a non-glacierized cell. When snow is transported onto a glacierized cell, it is becomes part of the snow cover on the glacier: it can thus reduce ice melt and become part of the glacier accumulation (Kuhn, 2003; Freudiger et al., 2017)."** (L279-285)*

Freudiger, D., Kohn, I., Seibert, J., Stahl, K., & Weiler, M. (2017). Snow redistribution for the hydrological modeling of alpine catchments. *Wiley Interdisciplinary Reviews: Water*, 4(5), e1232.
Kuhn, M. (2003). Redistribution of snow and glacier mass balance from a hydrometeorological model. Journal of Hydrology, 282(1-4), 95-103.

Line 268: "The glacier ice reservoir only decreases when…"
**Response:** Thank you for your suggestion, which we have implemented in the text:

*"**The glacier ice reservoir only decreases by melting** when it is not covered by snow, following a simple temperature-index scheme …" (L292-293)*

Line 288-295: This is also confusing and needs clarification. Is the Huss et al relationship applied to the distribution of elevations from combining all the rasterized glacier cells across the domain? Or do you group individual rasterized glacier cells as belonging to specific real-world glaciers based on where they are located? And then you use the distribution of model elevations associated with those real-life groupings? Otherwise wouldn't each rasterized glacier cell have an elevation change in direct correspondence to its mass balance change?

**Response:** Thank you for making us aware of the need for further clarification of the delta h parameterization. We have clarified that we do indeed apply this on a glacier by glacier basis.

*"Before running the model, **we assign a glacier ID to each group of glacier cells that are part of an individual glaciers based on the RGI outlines. Then, we** create offline maps of distributed glacier thickness, where each individual glacier loses a specific fraction of **its** mass (in steps of 1 percent mass loss) following Seibert et al. (2018a)." (L310-313)*

*"Mass changes do not necessarily occur in steps of 1 percent, leading to leftover mass or mass loss: for example, if **the** total mass loss **of a glacier** is 2.3 percent of the initial volume, we are left with 0.3 percent of leftover mass loss **for this glacier**. To address this, we distribute such leftover mass or mass loss evenly over the glacier area." (L315-318)*

Line 303-308: I would describe this as a sensitivity experiment. Do both the thickness of the upper soil layer and the total soil thickness vary spatially in the model? When you state that your sensitivity test is to halve the upper layer thickness it sounds like it varies over the region, but then when you state the maximum upper layer thickness it sounds like it is spatially uniform.

**Response:** Thank you for your suggestion. While the different model experiments we performed

to find a suitable change can be seen as a sensitivity analysis, we do not consider the entire change to the soil thickness to be a sensitivity experiment. Our aim is not to test how sensitive the model is to changes in the model architecture, but to apply a targeted and deliberate change to the soil thickness to make it more realistic. In our opinion, this is more analogous to manual calibration than to a sensitivity experiment. We have clarified that the upper layer thickness is indeed uniform over the domain.

*"Based on **a sensitivity analysis**, we decided to reduce the size of the top soil layer by making it half as thick everywhere, while maintaining the total soil thickness constant. Since the  depth of the upper soil layer in PCR-GLOBWB 2.0 **is constant over the domain** and by default set to 30 cm (Bierkens and Van Beek, 2009), halving the thickness still corresponds to a thickness of 15 cm, which is in line with the range of thicknesses that other LHMs are able to resolve (Telteu et al., 2021)."* (L328-331)

Lines 369: afterwards this is referred to as the "water gap", so please put this in parenthesis somewhere here.
**Response:** Thank you for making us aware of this. In response to the comments of another reviewer, we have decided to use the terminology provided by the original paper of Salwey et al., 2023, namely water balance signature (WB). The "water gap" would then be referred to as "negative value of WB". We have implemented this in the text.

*"**Another way** to separate regulated from natural catchments **is the water balance signature (WB), which describes** the normalized deviation from a closed water balance assuming no long-term storage effects (Salwey et al., 2023)."* (L400-402)

Line 414-417: Check references to figures and forcings. I think there is a mistake here where either Figure 4H should read 4I or one of the references to the forcings should read CHELSA instead of CERRA-CHELSA.
**Response:** Thank you for pointing out this inconsistency. We have changed the references to the Figures.

Lines ~425/Figure 5: It might be helpful to explicitly state that the large amount of SWE present during the summer in the benchmark (at high elevations) and ERA5 data (at middle and high elevations) is due to the presence of snow towers and that the inclusion of snow transport (present in the runs labelled "transport", "uncalibrated", and "full run") removes this unphysical effect. Also, please discuss the differences in SWE magnitude between the LHM model versions and the CERRA-Land analysis – does it also have snow build up in some cells, but resets to near-zero every year?
**Response:** Thank you for pointing out the need for further clarification. We have highlighted in the text that snow transport is the reason why these snow towers are removed and that ERA5-Land and CERRA-Land also suffer from snow build up.

*"..., where **unrealistic** snow towers were a major issue (compare the Transport run with the CERRA-CHELSA benchmark run in Figure 5 C and F). **Here, the snow transport scheme ensures that the snow is redistributed to lower elevations, where it subsequently melts away. ERA5-Land and CERRA-Land also show very high SWE values suggesting that these models also suffer from unrealistic snow build-up.**"* (L465-469)

Figure 5: This is a really small point but your vertical axis starts a zero in plots a and d but below zero in plots b,c,e,f.

**Response:** Thank you for the detailed look at our figures. We have changed them so the axis starts at 0 everywhere.

Line 431: There is no calibrated snow run labelled in the figure. Are you using the Full run as a proxy for it?

**Response:** Thank you for pointing out this inconsistency. Indeed, we use the Full model run as a proxy for Y as it includes the same snow set-up. We have changed this here in the text and have further highlighted this in the Figure caption.

*"....and the SNOWGRID product for Austria.* **Note that for simplicity we only show the Full model run instead of the Calibrated snow run, as these have the same snow module configuration.***" (caption Figure 5)*

Line 442-443: I don't think I agree with this conclusion. Based on Fig 7a there doesn't seem to be much correlation with water gap sign and I don't see a pattern of KGESS associated with either + or o in plots b,c. I do see a correlation of increased (decreased) KGE skill at locations with higher (lower) snowfall fraction with perhaps a weak dependence on glacier fraction. (The connection between performance and water gap sign for the snow/glacier modules and soil change are much more apparent).

**Response:** Thank you for highlighting this. We acknowledge that the KGESS mostly improves for catchments with high snowfall fractions. Our conclusions regarding the WB signature or reservoirs are mostly valid for such snow-dominated catchments. In the text, we have tried to better explain what we meant, limited the generalizability of the conclusions and do acknowledge that the relationship is weaker than for the other variables.

*"Figure 7A, B, and C illustrate that the* **changes in the snow module** *mainly* **improve** *model performance for discharge* **in catchments with high snowfall fractions, whereas in catchments with low snowfall fractions the changes are negligible or slightly negative. Still, some catchments with high snowfall fractions experience decreases in performance for discharge: these decreases mostly happen in the presence of reservoirs and/or negative values of WB ...***" (L491-494)*

Figure 6: This is a useful figure and tracks the progression of alterations nicely. Based on this, I'd suggest moving the KGE plot below plots a-f so that the full figure can take up more width on the page.

**Response:** We have changed the Figure accordingly.

Figure 7: I suggest labelling snow fraction as "snowfall fraction".

**Response:** Thank you for the suggestion, which we have implemented in the Figures and in the text.

Figure 8: I don't find plot 8i helpful/insightful as currently presented and discussed. It would be fine to remove, retain the stated numbers in the text regarding the equilibrium experiment and just leave the model-obs comparisons as shown in plots 8g,h.

**Response:** Thank you for indicating the need to improve this Figure. Based on this comment and comments of other reviewers, we have made significant changes to Figure 8 to improve the legibility of the figure. We have removed subplot 8I from the Figure and given more space to 8G and H.

Figure 8: The glacier outlines really obscure the elevation change results. Is it possible to improve on these maps? Perhaps it's possible to use grey in all four maps to represent non-glaciated regions and to use a color palette for the elevation change that goes through white at zero instead of grey?

**Response:** We have made significant changes to Figure 8 and removed the glacier outlines of smaller glaciers to improve the readability of the figure.

Figure 8: I suggest removing the sentence "Finally, the response of glaciers to continuous forcing with the mean mass balance from 1990–2018 (see Section 2.4)." from the caption.

**Response:** Thank you for the suggestion, which we have implemented in the caption. We did retain the time period indication and added it later in the caption.

*"(H) Comparison of the evolution of glacier volume over time under this **continuous forcing with their mean mass balance from 1990-2018** to modelled estimates from Zekollari et al., (2020)." (Caption Figure 8)*

Line 461: You subsequently define rainfall-dominated catchments to be Ps/P < 30% and improvements in Fig 7e are all for snowfall fractions higher than this value.
**Response:** Thank you pointing out this inconsistency. We have changed this in the text.

*"The positive effect of glaciers is much less visible in catchments with a small snow fraction,  (see Figure 7E)." (L509-510)*

Line 491: Are the KGESS color bars the same in Fig 11 and 4? If so the changes in skill between non-calibrated and calibrated periods appear smaller on average than the effect of different forcing data (although the spread in values they cover at the extremes is close)
**Response:** Thank you for pointing this out. We have checked this and it is indeed the case that the difference in model performance when using different forcing datasets is larger than the change between the calibration and evaluation period. Note however that most of the model is not calibrated. However, as we have moved this Figure to the Supporting Information along with most of the associated text based on comments from other reviewers, we have not highlighted this in the text.

Figure 11: It's hard to distinguish the blue and black colors used, particularly in the legend. Try a more differentiated color choice.
**Response:** Thank you for pointing this out. We have changed the colors in the Figure (which was moved to the Supporting Information).

Line 505: Your conclusion that "The representation of soil moisture ... needs to be improved in LHMs": This might be true but I would argue your results also suggest there is a need to compare observed estimates of soil moisture and simulated values in a more representative manner. (I don't think you need to do this yourself in this paper.)
**Response:** Thank you for pointing out the need for a more representative comparison. We added this call to the text.

*"Please note that the reference product used for computing model errors can have its own biases (Dorigo et al., 2015, 2017) **and has a much coarser spatial resolution than our model, so** error estimates might not be entirely representative. **Our results therefore suggest that there is a need for more representative ways to compare soil moisture simulations and observations. Still,** the representation of soil moisture and fast discharge responses needs to be further improved if LHMs are supposed to be applicable at smaller spatial scales." (L545-549)*

Figure 12: I suggest removing the label 'B.' from the first plot as it looks like a location you will refer to afterwards. Instead specify in the caption: Swiss canton of Grisons (inset shown in plot A)
**Response:** Thank you for your suggestion, which we have implemented accordingly in the figure.

Line 618: It might be worth specifying that these improvements apply even without calibration.
**Response:** We have added the following statement to the text:

*"Our analysis shows that our higher resolution model setup outperforms coarser reanalysis products such as CERRA-Land and ERA5-Land**, even without calibration** (Figure 5), but does*

*not quite reach performance of national-scale reanalysis products such as the OSHD product."*
*(L659-661)*

Lines 630: I think this claim would require additional testing. You only test about 0.5 degrees away from your calibration period temperature (local increase) but for a climate change study you'd probably want to model a global mean temperature increase of another 2 degrees (and more locally). This would push your model quite a bit further than you've tested.

**Response:** Thank you for pointing out the need for further testing to support this claim. We acknowledge that the temperature changes over the study period are limited and agree that the statement here is too strong.
Furthermore, we have moved the Figure to the Supporting Information, as it is not the main focus of this paper and the removal simplified the structure of the text.

Textual changes:

*"**A short** evaluation, which explicitly addressed model transferability, shows that model performance for discharge and SWE remains mostly consistent over the warmer and colder evaluation periods compared to the reference period**, although the temperature changes over the study period are limited** (Figure **S1** in the **Supporting Information**)." (L667-669)*

*Old: "We therefore have some confidence in the transferability of our model to warmer climate conditions. Still, the general caveats the model remain applicable ..."*
*New: "**All of this suggests a reduced sensitivity of our model to increases in temperature.** Still, the general caveats the model remain applicable ..." (L672-673)*

Line 656: I don't think it's fair to consider the CERRA-CHELSA forcing dynamically downscaled for the given model setup since the precip (likely the most important control) is taken as is from CERRA-Land at 5.5km and this is substantially coarser than your model at ~1km. Whereas the other products were downscaled (statistically) to 30". The way it is currently worded it sounds like the expectation was dynamical downscaling should yield improvements over statistical downscaling but I don't think you tested this hypothesis fairly (at similar resolutions) in your setup. I think you can still conclude that the choice of precip forcing did not make as large a difference on the resulting discharge accuracy as one might naively expect given the higher correlation and lower bias of the CERRA-CHELSA precip with observations). I think the conclusions as worded at lines 598-601 are more consistent with your experiments.

**Response:** Thank you for raising this point. While it is correct that the precipitation product is not further downscaled from CERRA-Land to the model resolution, we argue that the products should be placed in the context of the model dynamics of the underlying products.
Whereas products like ERA5 (or W5E5) underlying the STANDARD or CHELSA products have resolutions of 31 km or more, the dynamics in CERRA and CERRA-Land are run at a resolution of 5.5 km (Ridal et al., 2024), much closer to the resolution of our model. We thus hypothesized that CERRA should be able to represent small scale dynamics at a much higher resolution than the other products, even if these other products are statistically downscaled to the model resolution.

That being said, we agree that the wording can be improved: we removed the phrase dynamically-downscaled as we think this might indeed lead to the assumption that the data were downscaled to the model resolution and we named the underlying resolutions more explicitly.

In the introduction:

*"(H1) using  forcing products that include a representation of smaller-scale atmospheric dynamics compared to other forcing products," (L142-143)*

In the conclusion:

*"Discharge simulations forced by a **reanalysis product using high-resolution atmospheric dynamics at 5.5 km** (CERRA-CHELSA) did not consistently outperform runs with the other forcing products **which use coarser atmospheric dynamics at 31 km** (STANDARD and CHELSA; Hypothesis 1 is not supported)." (L695-697)*

Ridal, M., Bazile, E., Le Moigne, P., Randriamampianina, R., Schimanke, S., Andrae, U., ... & Wang, Z. Q. (2024). Cerra, the Copernicus European Regional ReAnalysis system. Quarterly Journal of the Royal Meteorological Society, 150(763), 3385-3411.

663-664: The final sentence is really vaguely worded. Omit or add some more specificity on the types of questions you think the model is ideally suited for.

**Response:** Thank you for pointing this out. We rephrased this sentence and highlighted possible applications.

*"Finally, we presented a new model setup with an improved representation of hydrological processes relevant in alpine regions, which is well suited to study regional and larger-scale streamflow and snow patterns in and around mountain regions. This new setup can be used to **help quantify water resources and to study how these are impacted by human water use or climate in the Alpine region or** around the world." (L707-710)*

Line 701: 50 meters? Specify that E_max and E_min are measured in meters.

**Response:** Thank you for pointing out the missing units, we have added these to the text:

*"Each glacier has a minimum surface elevation $E_{min}$ and a maximum surface elevation $E_{max}$ **(both in meters above sea level)**. This topographic range of the glacier surface can be split into N elevation zones (in our case into 20 steps, with $E_{min}$ and $E_{max}$ rounded to the nearest multiple of 50 m)." (L745-747)*

Line 710: What is lower-case m? the units of total glacier mass loss specified in units of meters? To me, writing (m) reads like "is a function of the variable *m*". I suggest rewriting to avoid this with something along the lines of: "This is done by means of a scaling factor f_S. This scaling factor is the ratio between the total mass loss over the glacier ΔM (units of m. water equivalent) and the integrated normalized change in surface elevation, ...) The units of all the subsequent variables should be clear from specifying those of ΔM.

**Response:** Thank you for your suggestion. We have implemented it in the text. However, we still maintained the specification that $h_i$ is the ice thickness also in units of m, water equivalent to facilitate orientation of the reader.

*"This is done by means of a scaling factor $f_s$. This scaling factor is the ratio between the total mass loss over the glacier ΔM **(units of m water equivalent)** and the integrated normalized change in surface elevation, scaled by the surface area of the elevation zone $A_i$ (as a fraction of the total glacier area)." (L755-759)*

*"...and h_i is the ice thickness **(units of** m water equivalent) in each cell in that specific elevation zone." (L761)*

**Technical edits:**

Line 136: downscaled
**Response:** Based on one of your earlier comments, we have decided to remove this word from the text.

Line 554: "than that of our LHM" or "than the resolution simulated here"
**Response:** We have changed this in the text.

---

## Author Response (AR2)

Dear Dr. Hendricks Franssen,

Thank you very much for accepting our manuscript for publication.

Following the comment of Reviewer 2, we have made the following technical correction:

Original (L147-148): *'[…] by calibrating SWE against a detailed regional SWE reanalysis […]'.*

Updated (L147-148) : *'[…] by calibrating snow module parameters against a detailed regional SWE reanalysis […]'*

Furthermore, we have added a missing affiliation to one of the co-authors *(Swiss Federal Institute for Forest, Snow and Landscape Research WSL* in Birmensdorf*)*.

Sincerely,

Joren Janzing on behalf of the authors